# Low-Dimensional Structure in the Space of Language Representations is Reflected in Brain Responses

**Richard Antonello**
UT Austin
rjantonello@utexas.edu

**Javier Turek**
Intel Labs
javier.turek@intel.com

**Vy Vo**
Intel Labs
vy.vo@intel.com

**Alexander Huth**
UT Austin
huth@cs.utexas.edu

## Abstract

How related are the representations learned by neural language models, translation models, and language tagging tasks? We answer this question by adapting an encoder-decoder transfer learning method from computer vision to investigate the structure among 100 different feature spaces extracted from hidden representations of various networks trained on language tasks. This method reveals a low-dimensional structure where language models and translation models smoothly interpolate between word embeddings, syntactic and semantic tasks, and future word embeddings. We call this low-dimensional structure a *language representation embedding* because it encodes the relationships between representations needed to process language for a variety of NLP (natural language processing) tasks. We find that this representation embedding can predict how well each individual feature space maps to human brain responses to natural language stimuli recorded using fMRI. Additionally, we find that the principal dimension of this structure can be used to create a metric which highlights the brain's natural language processing hierarchy. This suggests that the embedding captures some part of the brain's natural language representation structure.

## 1   Introduction

There are a multitude of common techniques for analytically representing the information contained in natural language. At the word level, language is often represented by word embeddings, which capture some aspects of word meaning using word co-occurrence statistics [8, 25]. Language representations that highlight specific linguistic properties, such as parts-of-speech [29] or sentence chunks [1], or that utilize well-known NLP models such as the intermediate layers of pretrained language models [6, 10, 21, 27], are also frequently studied [12, 31]. In fields such as linguistics, natural language processing, and cognitive neuroscience, qualitative adjectives are often used to describe these language representations – e.g. "low-level" or "high-level" and "syntactic" or "semantic". The use of these words belies an unstated hypothesis about the nature of the space of language representations – namely that this space is fundamentally low-dimensional, and therefore that the information from the representations in this space can be efficiently described using a few categorical descriptors. In this work, we attempt to directly map the low-dimensional space of language representations by generating "representation embeddings" using a method inspired by the work of Zamir et al. [44]. This method uses the *transfer properties between representations* to map their relationships.

The work described here has two main contributions. First, we used the representation embeddings to demonstrate the existence of low-dimensional structure within the space of language representations.

35th Conference on Neural Information Processing Systems (NeurIPS 2021).

We then used this structure to explore the relationships between – and gain deeper insight about – frequently used language representations. How do the intermediate layers of prominent language models relate to one another? How do the abstractions used by these layers evolve from low-level word embeddings of a context to a representation of the predicted next word for that context? Do different language models follow similar representation patterns? What are the differences between how unidirectional and bidirectional language models represent information? These are examples of the types of questions we can explore utilizing our representation embedding space. Second, we showed that this low-dimensional structure is reflected in brain responses predicted by these representation embeddings. In particular, we show that mapping the principal dimension of the representation embeddings onto the brain recovers, broadly, known language processing hierarchies. We also show that the representation embeddings can be used to predict which representations map well to each area in the brain.

## 2  Related Work

Our work closely follows the methods developed by Zamir et al. [44]. Those authors generated a taxonomy of vision tasks and analyzed their relationships in that space. This was achieved by applying transfer learning across vision tasks, and showed that this can reduce data labeling requirements. We used some of their methods, such as transfer modeling and normalization using an analytic hierarchy process (AHP) [28], for our analysis on language tasks. Following the same lines but for NLP, the work from Vu et al. [37] explores transferability across NLP tasks. Those authors focus on the application of transfer learning to show that the amount of labeled data, source and target tasks, and domain similarity are all relevant factors for the quality of transferability. Kornblith et al. [20] investigated and compared techniques for measuring the similarity of neural network representations, such as canonical correlation analysis (CCA) and centered kernel alignment (CKA).

Our analysis of the relationship of between the language representation space and the brain leans heavily on the concept of "encoding models". Recently, encoding models have been widely applied as predictive models of brain responses to natural language [17, 40]. Jain and Huth [18] applied encoding models to representations of an LSTM language model that incorporates context into neural representations. These representations are explored for different timescales of language processing in Jain et al. [19]. Toneva and Wehbe [33] explored the encoding model performance of different language model layers to improve neural network performance. Caucheteux and King [5] observed differences in the convergence of hidden state representations from ANNs trained on various visual and language processing tasks to brain-like representations. Schrimpf et al. [30] systematically examined the performance of encoding models across a variety of representations and neural response datasets and observed consistent high performance of Transformer-based model hidden states at predicting neural responses to language. Wang et al. [39] constructed encoding models on a set of 21 vision tasks from Zamir et al. [44] with the objective of localizing tasks to specific brain regions. Those authors built a task graph of brain regions showing similar representation for highly transferable tasks.

## 3  Methods Overview

Our method aims to use the transfer properties between different language representations to generate quantitative descriptions of their relationships. These descriptions can then be used as an embedding space without labelling any individual representation with predefined notions of what that representation is or how it relates to other representations. This is analogous to how word embeddings use co-occurrence statistics to generate quantitative descriptions of each word without labelling any individual word with our predefined notions of what that word means. By using transfer properties to generate the embedding for each representation, we implicitly assume that language representations with similar transfer properties are themselves similar.

We define a *representation* $t$ as a function over a language input that extracts a vector of meaningful features for each word. Some representations can be computed independently for each word, such as word embeddings [1, 10, 25], while others depend on both a word and its context, such as part of speech labels [1], or the hidden state in some intermediate layer of a language model [5, 6, 27, 30]. We define a stimulus $s$ as a string of words drawn from some natural language source, and $\mathcal{S}$ the set of all stimuli from that source. To generate a representation embedding over a set of representations $\mathcal{T}$ that are derivable from the same stimuli $\mathcal{S}$ we will construct time-dependent mappings between

representations. For a representation $t \in \mathcal{T}$, and for all times $j$ in our stimulus $\mathbf{s} \in \mathcal{S}$, the vector $t(\mathbf{s}_j)$ is defined. Next, for each pair of representations $t_1, t_2 \in \mathcal{T}$ we will attempt to map $t_1(\mathbf{s}_j)$ to $t_2(\mathbf{s}_j)$. This provides a transfer mapping $f_{t_1 \to t_2}(\cdot)$ that can be learned for each pair of representations,

$$f_{t_1 \to t_2}(t_1(\mathbf{s}_j)) \approx t_2(\mathbf{s}_j) \qquad \forall j.$$

However, learning these transfer mappings directly would make comparison across representations difficult, as many of the representations have different dimensionalities. Instead, we adapted the encoder-decoder framework proposed in Zamir et al. [44], which ensures that every transfer mapping has the same input dimensionality. This method proceeds in three parts (Figure 1):

First, we train a linear encoder for each representation that compresses the information in the input down into a latent space. The latent space should only contain information necessary to predict the given representation's features from our input. This is accomplished by training an encoder-decoder from the input to the given representation via a latent space, then discarding the decoder.

Second, we train a new decoder for each pair of representations that uses the previously generated latent space for one representation to predict the other representation.

Third, we evaluate the performance of each decoder targeting a given representation relative to every other decoder targeting that same representation. This provides a measure of how much of the information in one representation is present in every other representation. Just as words that have similar co-occurrence statistics—when measured against a large enough group of other words—are thought to have similar meanings, we argue that pairs of representations that have similar transfer properties—when measured against a large enough group of other representations—have similar information content.

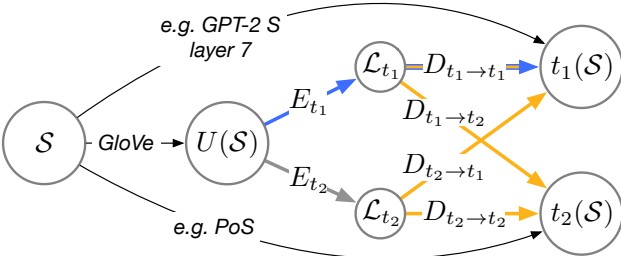

Figure 1: The encoder-decoder strategy used in our method, adapted from Zamir et al. [44]. $\mathcal{S}$ is the natural language stimuli. We chose to represent stimuli in the universal input feature space $U(\mathcal{S})$ as GloVe word embeddings. Encoders $E_{t_i}$ were trained using a bottlenecked linear encoder-decoder network, which outputs to $t_i(\mathcal{S})$ (*blue arrows*). The decoding half of this network was then discarded, and the encoding half used to generate a latent space $\mathcal{L}_{t_i}$ for each representation $t_i$. Then, a decoder $D_{t_i \to t_j}$ is trained from each latent space $i$ to each representation $j$ (*orange arrows*). The performance of decoders that map to the same final representation are then compared to one another.

### 3.1 Generating Representation Encoders

We define[1] $U(\mathcal{S})$ as the universal input feature space for our stimuli $\mathcal{S}$. In practice, this space should represent the input with high fidelity, so could be selected as a one-hot encoding of each input token, or word embeddings for each input token. We use GloVe word embeddings for $U(\mathcal{S})$. Now let $\mathcal{T}$ be a set of representations over $\mathcal{S}$. For each representation $t \in \mathcal{T}$, we generate an encoder $E_t(\cdot)$ such that the encoder extracts only information in $U(\mathcal{S})$ that is needed to predict $t(\mathcal{S})$. We do this by using a bottlenecked linear neural network that maps every $\mathbf{u} \in U(\mathcal{S})$ to an intermediate low-dimensional latent space $\mathcal{L}_t = E_t(U(\mathcal{S}))$ and then maps it to the given representation space,

$$t(\mathbf{s}) \approx f(E_t(\mathbf{u})) \qquad \forall \mathbf{s} \in \mathcal{S} \wedge \mathbf{u} = U(\mathbf{s}),$$

where $f(\cdot)$ is mapping from $\mathcal{L}_t$ to $t(\mathcal{S})$. We used a small latent space of 20 dimensions to encourage the encoder to extract only the information in $U(\mathcal{S})$ that is relevant to compute $t(\mathcal{S})$. Experimentation showed that variations in the latent space size do not meaningfully change encoder performance.

---

[1]We assume that elements $j$ in a string $\mathbf{s}_j$ are properly treated, such that we can exclude the positional indexes for all $\mathbf{s}$. Thus, we define $U(\mathcal{S}) = \{U(\mathbf{s}) | \mathbf{s} \in \mathcal{S}\}$ as the result of applying a function $U(\cdot)$ to all elements $\mathbf{s} \in \mathcal{S}$.

Once we have learned these mappings, we assign $E_t$ to be our representation encoder for each representation $t \in \mathcal{T}$. Regardless of the dimensionality of the representation, each latent space has a fixed dimensionality, which enables a fair comparison between representations.

## 3.2 Generating Representation Decoders

The encoders for each representation generate a latent space $\mathcal{L}_t$ that extracts the information in $U(\mathcal{S})$ relevant to computing $t(\mathcal{S})$, while compressing away irrelevant information. For every pair of representations $(t_1, t_2) \in \mathcal{T}$, we next generate a decoder $D_{t_1 \to t_2}$ such that $D_{t_1 \to t_2}(\mathcal{L}_{t_1}) = D_{t_1 \to t_2}(E_{t_1}(U(\mathcal{S})))$ approximates $t_2(\mathcal{S})$. This yields a total of $n^2$ decoders, where $n = |\mathcal{T}|$ is the total number of representations. All networks were trained with batches of size 1024 and standard stochastic gradient descent with a learning rate of $10^{-4}$ for the initial encoders and $2 \times 10^{-5}$ for the decoders. Hyperparameters were chosen via coordinate descent.

## 3.3 Generating the Representation Embedding Matrix

Finally, we compare the performance of the $n$ decoders for each representation in order to generate the representation embedding space. To ensure that comparisons are made on equal footing, we only compare decoders with the same output representation (e.g. $D_{t_1 \to t_2}$ is compared with $D_{t_3 \to t_2}$, $D_{t_4 \to t_2}$, etc.). This is critical, because evaluating representations with varying dimensionalities would unfairly bias the results. This and related challenges were explored in detail in Zamir et al. [44].

We used the decoders to generate a pairwise *tournament matrix* $\mathbf{W}_t$ for each representation $t$ by "fighting" all pairs of decoders that output to representation $t$ using a held-out test set $\mathcal{S}_{test}$ of sentences. Element $(i, j)$ in $\mathbf{W}_t$ contains the ratio of samples in the test set for which $D_{t_i \to t}$ has lower MSE than $D_{t_j \to t}$, i.e., $\mathbf{W}_{t_{(i,j)}} = \frac{\mathbb{E}_{\mathbf{s} \in \mathcal{S}_{test}}\left[D_{t_i \to t}(\mathbf{s}) < D_{t_j \to t}(\mathbf{s})\right]}{\mathbb{E}_{\mathbf{s} \in \mathcal{S}_{test}}\left[D_{t_i \to t}(\mathbf{s}) > D_{t_j \to t}(\mathbf{s})\right]}$. For example, if the decoder $D_{A \to C}$ has lower mean squared error than decoder $D_{B \to C}$ for 75% of the data in $\mathcal{S}_{test}$, we assign the ratio of $0.75/0.25 = 3$ to entry $(A, B)$ in the tournament matrix $\mathbf{W}_C$ for representation $C$.

We then use this set of pairwise comparisons to approximate a total order over the quality of all encoders for each decoded representation. This was done using the Analytic Hierarchy Process (AHP) [28], a technique commonly used in operations research to convert many pairwise comparisons into an estimated total order. AHP establishes that the elements of the principal eigenvector of $\mathbf{W}_t$ constitutes a good candidate for a total order over the encoders for $t$. This eigenvector is proportional to the time that an infinite length random walk on the weighted bidirected graph induced by $\mathbf{W}_t$ will spend at any given representation. Thus, if the encoder for one representation is better than the others, the weight on that representation will be higher as compared to the weights of other encoders. This eigenvector is then normalized to sum to 1. This procedure yields a length $n$ vector for each of the $n$ representations. We set a value of $0.1$ into each vector at the position corresponding to the target representation, and then stack the resulting length $n$ vectors together into the representation embedding matrix, $\mathbf{R}$.

# 4 Results

## 4.1 Language Representation Embeddings

We applied our method to a set of 100 language representations taken from precomputed word embeddings or pretrained neural networks for NLP tasks operating on English language inputs.[2] We extracted representations from numerous different tasks, which included three word embedding spaces (GloVe, BERT-E, FLAIR) [25, 10, 1], three unidirectional language models (GPT-2 Small, GPT-2 Medium, Transformer-XL) [27, 6, 41], two masked bidirectional language models (BERT, ALBERT) [10, 21], four common interpretable language tagging tasks (named entity recognition, part-of-speech identification, sentence chunking, frame semantic parsing) [1], and two machine translation models (English → Mandarin, English → German) [32]. We also included the GloVe embedding for the next word in the sequence, which constitutes a low-dimensional representation of the ideal output of a language model. For multilayered networks (language and translation models), we extracted all intermediate layers as separate representations. Full descriptions of all representations, as well as details on how each feature space is extracted, can be found in Appendix A.

---

[2]Data and code for this paper are available at `https://github.com/HuthLab/rep_structure`.

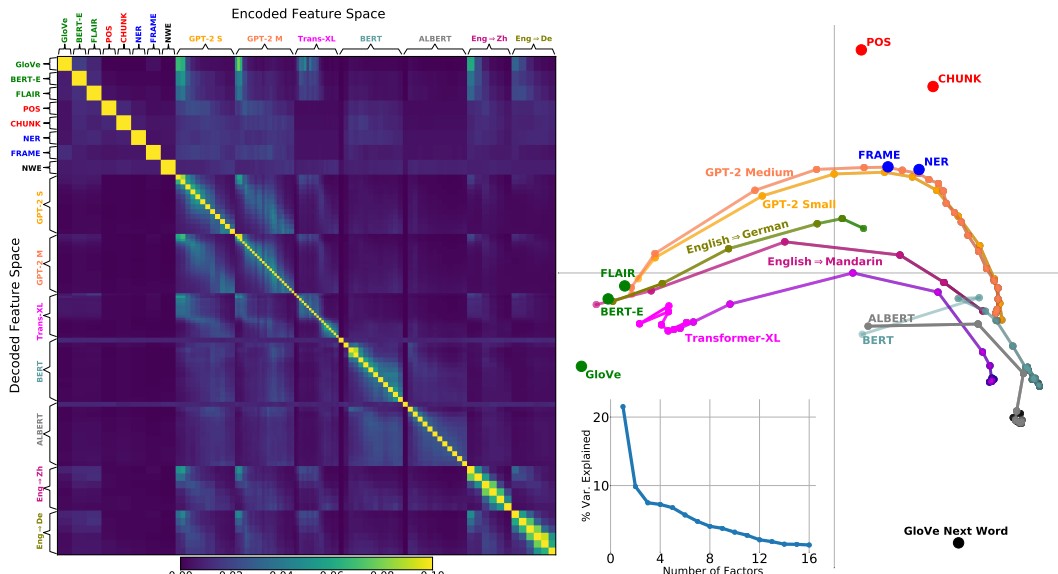

Figure 2: *Language Representation Embeddings with Low Dimensionality:* (*Left*): The representation embedding matrix **R** shows how well a given linguistic feature space (encoder, columns) transfers to another feature space (decoder, rows). For better visualization, rows and columns corresponding to different layers from the same network have been scaled down in this plot. A full-scale matrix is in the supplementary material. (*Right*): Applying multi-dimensional scaling to the representation embedding matrix reveals low-dimensional structure in the linguistic feature spaces. It is dominated by a left-to-right progression from the input word embedding, to syntactic and semantic tagging tasks near the middle layers of language models, to the next word embedding. Multidimensional scaling was weighted such that each full model had equal weight, ensuring that language models were not more influential on account of having more layers. The dominant main diagonal was set to 0.1 to preserve the effects of off-diagonal values. The scree plot in the lower left shows that these first two dimensions explain substantially more variance (22% and 10%) than other dimensions, demonstrating that the structure in this space is low-dimensional.

Applying our method to these representations, we generated a representation embedding space (Figure 2). Generating this space took roughly one week of compute time on a cluster of 53 GPUs (a combination of Titan X, Titan V, and Quadro RTXs) and 64 CPU servers of different characteristics (Intel[3] Broadwell, Skylake, and Cascade Lake). The representations were trained on a text corpus of stories from *The Moth Radio Hour* [36] which totalled approximately 54,000 words. The held-out test set for the tournament matrix included a story of 1,839 words. Each row shows the relative performance of each of the 99 other representations in decoding a given representation. Each column shows how well a given encoded representation can decode to the other representations. Note that there is a distinct asymmetry: if a given representation (e.g. part-of-speech tagging) is never the best encoder for any other representation, then its column will contain very low values. However, other representations may decode to this one quite well, so the corresponding part-of-speech row can contain higher values.

### 4.1.1 Multidimensional Scaling Analysis

To better visualize the relationships between the representations, we performed multidimensional scaling (MDS) [34] on the rows of the representation embedding matrix. We then plotted the representations according to their locations on the first two MDS dimensions (Figure 2, right). This showed that the layers of each of the unidirectional language models form similar trajectories through the space, beginning near the word embeddings (*green*) and ending relatively close to the next-word embedding (*black*). This trajectory is monotonic on the first MDS dimension, but values on the second MDS dimension rise and then fall, with the highest values assigned to the middle layers (e.g. layer 5 of 12 in GPT-2 small). Language tagging tasks that require semantic information,

---

[3]Intel, the Intel logo, and other Intel marks are trademarks of Intel Corporation or its subsidiaries. Other names and brands may be claimed as the property of others.

such as framing and named entity recognition (*blue*), are close in space to the middle layers of the unidirectional language models. Language tagging tasks that required syntactic information, such as part-of-speech identification and sentence chunking (*red*), are similarly clustered.

We also observe several other interesting relationships. Like language models, the machine translation representations also begin near the word embeddings. However, translation to Mandarin (Eng→Zh) appears to align more closely to the unidirectional language models than translation from English to German. This may be because English and Mandarin are less closely related to one another, and could thus require deeper syntactic and semantic processing for successful translation.

We also observe that the bidirectional language model representations begin farther away from the input word embedding, closer to the middle of the unidirectional language models. However, they also end up at a similar point, closer to the next-word embedding. This may be a result of the procedure to extract a feature space from the intermediate layers of the bidirectional language models, which differed meaningfully from extracting the intermediate layers of the unidirectional language models. Since these networks are trained on a masked language modeling task, we extracted the hidden representation for the token immediately preceding the mask token.

To study the dimensionality of the space of language representations, we examined the variance explained by low-dimensional approximations of the embedding matrix. The scree plot in Figure 2 shows the result of applying exploratory factor analysis (EFA) [43] to the representation embedding matrix. We see that the first few dimensions explain much more variance in the matrix than the other dimensions. The first dimension alone constitutes 22% of the total variance. This demonstrates that the representation embedding matrix—and thus, we infer, the space of language representations—has low-dimensional structure.

## 4.2 Representation Embeddings Predict fMRI Encoding Model Performance

Next, we hypothesized that these language representation embeddings are a useful representation of how natural language is processed in the human brain. Since it is the only system truly capable of producing and understanding complex language, the human brain is an excellent litmus test to determine whether this low-dimensional embedding captures the underlying structure of linguistic information. To test this hypothesis, we used the representation embeddings generated above to predict how well each representation is able to predict fMRI data collected from the human brain as the subject is listening to natural language stories.

### 4.2.1 fMRI Data

We used functional magnetic resonance imaging (fMRI) data collected from 5 human subjects as they listened to English language podcast stories over Sensimetrics S14 headphones. Subjects were not asked to make any responses, but simply to listen attentively to the stories. For encoding model training, each subject listened to at approximately 5 hours of unique stories across 5 scanning sessions, yielding a total of 9,189 datapoints for each voxel across the whole brain. For model testing, the subjects listened to the same test story once in each session (i.e. 5 times). These responses were then averaged across repetitions. Training and test stimuli are listed in Appendix B.1. Functional signal-to-noise ratios in each voxel were computed using the mean-explainable variance method from Nishimoto et al. [24] on the repeated test data. Only voxels within 8 mm of the mid-cortical surface were analyzed, yielding roughly 90,000 voxels per subject.

MRI data were collected on a 3T Siemens Skyra scanner at the University of Texas at Austin Biomedical Imaging Center using a 64-channel Siemens volume coil. Functional scans were collected using a gradient echo EPI sequence with repetition time (TR) = 2.00 s, echo time (TE) = 30.8 ms, flip angle = 71°, multi-band factor (simultaneous multi-slice) = 2, voxel size = 2.6mm x 2.6mm x 2.6mm (slice thickness = 2.6mm), matrix size = 84x84, and field of view = 220 mm. Anatomical data were collected using a T1-weighted multi-echo MP-RAGE sequence with voxel size = 1mm x 1mm x 1mm following the Freesurfer [15] morphometry protocol.

Experiments were approved by the University of Texas at Austin IRB. All subjects gave written informed consent. Subjects were compensated for their time at a rate of $25 per hour, or $262 for the entire experiment. Compensation for the 5 subjects totaled $1260.

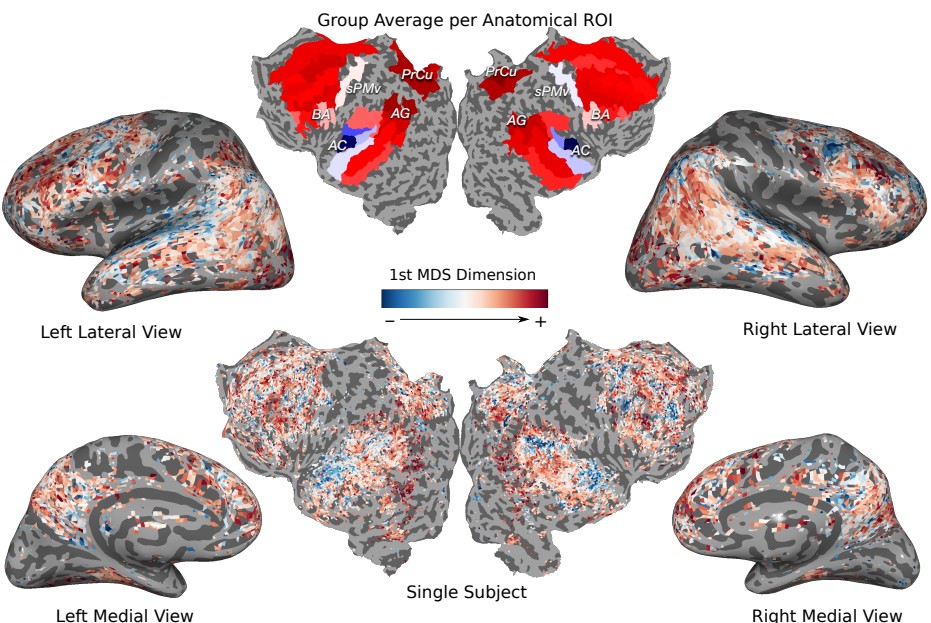

**Figure 3:** *Embedding Brain Voxels in the first MDS dimension:* Projection of the encoding performance vectors for each voxel in one subject (*lower center* flatmap and all 3D views) and averaged over all subjects within anatomical regions (*upper center* flatmap) over the 100 representations onto the first MDS dimension of the representation embeddings, which explains about 20% of the variance in the representation embeddings. Voxels with high values in this embedding (*red*) are better explained by representations that are more positive on the main MDS dimension (e.g. later language model layers), and voxels with low values (*blue*) are better explained by representations that are more negative (e.g. word embeddings). This dimension is notable as it is the main dimension along which language representations evolve from "earlier" representations such as word embeddings, to "later" representations such as intermediate layers in deep language models. Anatomical ROIs were defined automatically in each subject using Freesurfer with the Destrieux 2009 atlas [9]. Similar maps for the other subjects and a plot showing the numerical projection of regions in the MDS space are shown in Appendix D.1.

### 4.2.2 fMRI Encoding Models

Encoding models predict a measured brain response $B$, e.g. blood-oxygen-level dependent (BOLD) responses recorded with fMRI, based on the stimulus observed by the subject. Due to limitations on dataset size, encoding models are typically structured as linearized regression, where each regression predictor is some feature of the stimulus. Encoding models can provide insight into where and how information is represented in the brain. For instance, an encoding model that has a feature for whether a given auditory stimulus is a person's name can be used to determine which regions of the brain are activated by hearing names.

Here we constructed voxelwise encoding models using ridge regression for each of the 100 language representations $t$ analyzed above. Let $g(t_i)$ indicate a linearized ridge regression model that uses a temporally transformed version of the representation $t_i$ as predictors. The temporal transformation accounts for the lag in the hemodynamic response function [23, 17]. We use time delays of 2, 4, 6, and 8 seconds of the representation to generate this temporal transformation. For each subject $x$, voxel $v$, and representation $t_i$, we fit a separate encoding model to predict the BOLD response $\hat{B}$, i.e. $\hat{B}_{(x,v,t_i)} = g_{(x,v,t_i)}(t_i)$. An optimal ridge parameter $\alpha$ was estimated for each $g_{(x,v,t_i)}$ by using 50-fold Monte Carlo cross-validation, with a held-out validation set of 20% of the datapoints from the training dataset. We then measured encoding model performance $\rho$ by computing the correlation between the true and predicted BOLD responses for a separate test dataset consisting of one story. That is, $\rho_{(x,v,t_i)}$ measures the capacity of representation $t_i$ to explain the responses of voxel $v$ in subject $x$.

### 4.2.3 Mapping the First Dimension of the Representation Embedding Space to the Brain

If a brain area was specialized for processing a specific type of information, then we would expect representations that capture that information to be good predictors of that brain area. Each voxel in the brain can thus be thought of as another language representation, and we can infer where that representation would lie in the MDS space by projecting it onto those dimensions.

The main MDS dimension (left to right in Figure 2) is especially interesting as it seems to capture an intuitive notion of a language representation hierarchy. Representations with low values along the main dimension include word embeddings as well as the earliest layers of most of the language models and machine translation models. Representations with high values along the main dimension include the deeper layers of these models, as well as the majority of the interpretable syntactic and semantic representations. We tested whether this MDS dimension could capture patterns of hierarchical processing observed in cortex. First we defined $\mathbf{p}_{(x,v)} = \mathbf{zscore}([\rho_{(x,v,t_1)} \cdots \rho_{(x,v,t_n)}])$ as the 100-dimensional vector that denotes the encoding model performances for each subject $x$ and voxel $v$, z-scored over all representations $\mathcal{T}$. We then projected each $\mathbf{p}_{(x,v)}$ onto the first MDS dimension by taking their dot product. This gave us a value that quantifies which part of the first MDS dimension best predicts the activity of a given voxel.

Figure 3 shows shows the projection of each voxel for one subject (*lower center*) and averaged across subjects in each anatomical ROI (*upper center*) onto the first MDS dimension. Blue voxels and regions are better predicted by representations that are low on the first MDS dimension, whereas red voxels and regions are better predicted by representations that are high on the first MDS dimension.

While the literature has not settled on a single hierarchical view of language processing in human cortex, one point of general agreement among existing theories is that there is a set of "lower" language areas, including Wernicke's area/auditory cortex (AC), Broca's area (BA), and the premotor speech area sPMv [13, 16]. Across our five subjects, we see negative projections on the first PC (corresponding to earlier LM layers) in AC, but both positive and negative projections in BA and sPMv. This matches other results using narrative stories and encoding models [7] where it was found that, of these three core language areas, AC is the best explained by lower-level features (phonemes and sound spectrum).

Outside of the core language areas, the literature is more divided on representational hierarchy. It is broadly agreed upon that these other areas, including much of the temporal, parietal, and prefrontal cortex, constitute the "semantic system" [4] in which language meaning is derived and represented. Our data show that most of these regions have positive projections on the first PC, corresponding to later LM layers. More fine-grained analyses have suggested that some areas, such as the angular gyrus (AG) and precuneus (PrCu), contain the highest-level representations [19, 22]. This matches our group-level results, which show that the AG and PrCu have the most positive projections on the first PC of any brain areas.

### 4.2.4 Predicting Encoding Model Performance with Representation Embeddings

If the representation embeddings reflect how the brain represents linguistic information, then it should be possible to match each representation embedding $\mathbf{r}_k$ to its corresponding performance vector $\rho_{(x,\bullet,t_k)}$, which describes how well that representation can predict responses in each cortical voxel. We test this via a leave-two-out experiment, where we train a learner using 98 $(\mathbf{r}_k, \rho_{(x,\bullet,t_k)})$ pairs and then use this learner to predict which of the remaining two representation embeddings match the remaining two performance vectors. If the correlation between the predicted performance vector for representation A matches the ground truth performance vector for representation A better than the ground truth performance vector for representation B (and similarly B matches B better than it matches A), then we have succeeded in correctly discriminating a performance vector from its corresponding representation embedding.

For each subject $x$ and pair of language representations $(t_i, t_j)$, we trained a linear regression $\hat{h}_{(x,i,j)}$ to map a representation embedding $\mathbf{r}_k$ to the corresponding encoding model performance built from $t_k$ over all voxels.

$$\hat{h}_{(x,t_i,t_j)}(\mathbf{r}_k) \approx corr(g_{(x,\bullet,k)}(t_k), B_{(x,\bullet)}) = \rho_{(x,\bullet,t_k)},$$

$\mathbf{r}_k$ is defined as the 196-element vector that concatenates the row and column that correspond to the representation $t_i$, leaving out the four elements corresponding to the held out pair: $\mathbf{R_{ii}}$, $\mathbf{R_{ij}}$, $\mathbf{R_{ji}}$, and

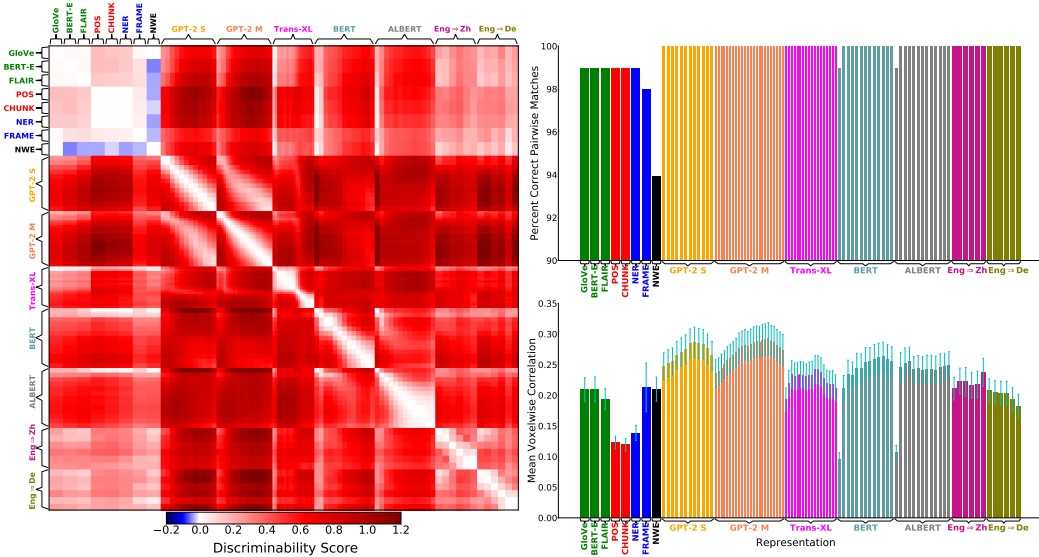

Figure 4: *Representation Embeddings Reflect Brain Responses*: (*Left*): Discriminability score matrix $\mathbf{M}$, the average across each subject matrix $\mathbf{M}_x$, which is computed as described in Section 4.2.4. For each representation, we fit and tested encoding models that predict the fMRI response in each cortical voxel, yielding a pattern of prediction performance across the brain. For each pair, we then tested whether the brain patterns could be correctly matched to the representations on the basis of the representation embeddings shown in Figure 2. Highly discriminable pairs appear red, non-discriminable pairs white, and pairs that are less discriminable than expected by chance appear blue. Most pairs of representations yield brain patterns that are easy to distinguish using the embeddings, suggesting that these embeddings reflect the structure of representations in the human brain. However, some pairs are similar enough in both embedding and brain that discrimination between them falls to chance level, such as the word embeddings (green labels) and nearby layers of language models. (*Upper Right*): The percentage of pairwise matches for each representation where the match is correct more often than not (on 3 or more subjects). Almost all representations can be correctly matched with their corresponding performance vector $\mathbf{p}$, with the interpretable representations being the most difficult to distinguish. (*Lower Right*): Mean voxelwise correlation for encoding models built using each representation of the natural language story stimuli. As seen in other literature [30], the intermediate layers of Transformer-based language models work best as encoding model representations.

$\mathbf{R_{jj}}$. The model $\hat{h}_{(x,t_i,t_j)}$ was then used to predict the held out encoding model performance across all voxels, $\rho(x, \bullet, t_i)$ and $\rho(x, \bullet, t_j)$. For each subject, we then computed discriminability scores for all pairs of representations to generate a matrix $\mathbf{M}_x$. This quantifies how much better the predicted encoding model performance for $t_i$ matched with the true encoding model performance for $t_i$ than the true encoding model performance for $t_j$ (and vice versa):

$$\mathbf{M}_{x_{(i,j)}} = (corr(\hat{h}_{(x,t_i,t_j)}(\mathbf{r}_i), \rho_{(x,\bullet,t_i)}) + (corr(\hat{h}_{(x,t_i,t_j)}(\mathbf{r}_j), \rho_{(x,\bullet,t_j)})))$$
$$- (corr(\hat{h}_{(x,t_i,t_j)}(\mathbf{r}_i), \rho_{(x,\bullet,t_j)}) + (corr(\hat{h}_{(x,t_i,t_j)}(\mathbf{r}_j), \rho_{(x,\bullet,t_i)})))$$

Figure 4 shows the discriminability scores for all representation pairs averaged across subjects, as well as the average encoding model performance for each individual representation. The vast majority of pairs have high discriminability scores (*red*), indicating that the representation embeddings broadly capture the structure of the brain's language representations.

However, a few pairs have negative scores (*blue*), indicating below-chance discrimination performance. This could happen by chance, or it could happen if the task affinities are very different from the brain representations. We believe the negative scores for GloVe NWE are caused by the latter. This representation is highly dissimilar from the others, hence its far-removed location in the MDS space. However, the temporal imprecision of the BOLD responses recorded by fMRI (on the order of 1 second, or 2-5 words) renders the GloVe NWE and normal GloVe embeddings nearly identical for predicting brain data. This disconnect is likely the cause of the negative scores.

We also computed how often the discriminability score was higher than 0 in the majority of subjects (at least 3 out of 5). We see (Figure 4, *upper right*) that encoding model performances are well predicted by the representation embeddings, with representation embeddings getting matched to their corresponding encoding model performances over 90% of the time for all representations, and 100% of the time for the majority of representations.

## 5 Future Work, Limitations, and Conclusions

By measuring transferability between different language representations, we were able to visualize the relationships between 100 representations extracted from common NLP tasks. This uncovered a low-dimensional structure that illustrates how different tasks are related to one another, and how representations evolve over depth in different language models. Finally, we show evidence that these representation embeddings capture the structure of language representations in the human brain.

There are further possible iterations of this method that may shed additional light on the structure of language representations in the brain. Since this method relies on using the transfer properties between representations to generate embeddings, it is necessary that the chosen set of representations spans the rich space of language representations. If some part of the space of language representations has not been sampled, then the representation embeddings may fail to capture some aspects of the structure of that space. We tried to ameliorate this concern by sampling a large number of representations (100). However, it is likely that these chosen representations still do not capture the full diversity of the space. In future work, an even larger number of representations could be used to better map this space.

In a similar vein, we selected GloVe embeddings as the "universal" input representation for our encoder-decoder setup. GloVe embeddings are likely not "universal" in the same way that a one-hot encoding of the story might be, but practical concerns made this approximation necessary. Use of other universal spaces could also be enlightening, such as directly using fMRI data to map to a latent representation space—i.e. replace $U(S)$ in Figure 1 with fMRI BOLD responses $B$ recorded from the same natural language experiment. Of course, the possibilities are not restricted to language representations, as previous work had focused on vision instead [39, 44].

We also used linear as opposed to nonlinear networks for our encoder-decoders. This limits the expressiveness of the transfer networks, and thus may under-estimate the relatedness between tasks. However, using only linear networks also increases interpretability by ensuring that only simply-related representations can transfer to one another, as also noted by others [2, 26]. Future work may explore the use of regularized nonlinear models for the encoder-decoder networks.

This method does require extracting representations from a neural network trained to accomplish some task. For many tasks used in linguistics, neuroscience, and cognitive psychology, there is no existing pre-trained network. However, training artificial neural networks on a task that is used to measure cognition in humans and animals is an emerging method that can shed light on how these tasks are accomplished [11, 35, 42], and provide standardized benchmarks for comparing between different computational cognitive models. Furthermore, we can use our proposed method to understand what representations are needed to accomplish different tasks, and to visualize the relationships between those representations. For example, the debate on whether lexico-semantic and syntactic representations are actually separable [14] centers on evidence from different tasks that claim to measure syntactic representations versus semantic ones. The claims about the tasks themselves are difficult to test, but methods like those presented in this work can directly quantify how much information is shared between representations needed to accomplish those tasks, either in artificial neural networks or in neuroimaging data. Our proposed method may even provide new evidence in other domains where overlap has been observed between neural representations elicited by ostensibly different cognitive tasks [3, 38].

In sum, we believe that our work can provide a template for investigating relationships between linguistic and cognitive representations in many different domains.

## Acknowledgements

We would like to thank Amanda LeBel for collecting the fMRI data and reconstructing cortical surfaces, Lauren Wagner for annotating the experimental stimuli, and the anonymous reviewers for their insights and suggestions. Funding for this work was provided by the Burroughs-Wellcome Fund

Career Award at the Scientific Interface (CASI), Intel Corporation, the Whitehall Foundation, and the Alfred P. Sloan Foundation. The authors declare no conflicts of interest.

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
