# A    Representation Descriptions

**GloVE** [25] is a 300-dimensional word embedding space. It is an dimensionality-representation representation of word-word co-occurrence statistics. GloVe embeddings were sourced from `http://nlp.stanford.edu/data/glove.6B.zip`

**BERT-E** [10] is a 3072-dimensional contextualized word embedding space extracted from BERT. We used the Flair NLP [1] implementation of BERT embeddings.

**FLAIR** [1] is a 4096-dimensional contextualized character level word embedding space. We used the Flair NLP implementation of this word embedding space.

**POS** is the 53-dimensional representation stored at the pre-softmaxed output logit layer of the pre-trained FLAIR [1] part-of-speech LSTM-based sentence tagger.

**CHUNK** is the 45-dimensional representation stored at the pre-softmaxed output logit layer of the pre-trained FLAIR [1] LSTM-based sentence chunker.

**NER** is the 76-dimensional representation stored at the pre-softmaxed output logit layer of the pre-trained FLAIR [1] LSTM-based named entity recognition sentence tagger.

**FRAME** is the 5196-dimensional representation stored at the pre-softmaxed output logit layer of the pre-trained FLAIR [1] LSTM-based semantic framing (verb disambiguation) tagger.

**NWE** is the **GloVe** representation, offset by one word in the future.

The FlairNLP repository which we used for the preceding represenations can be found here: `https://github.com/flairNLP/`

**GPT-2 Small** is a set of 12 representations, each 768-dimensional, built from each hidden state output from the 12 transformers that compose the GPT-2 Small unidirectional language model [27]. Representations were built using a sliding window of 64 words as a context. We used the HuggingFace [41] implementation of this network to extract feature for these representations.

**GPT-2 Medium** is a set of 24 representations, each 1024-dimensional, built from each hidden state output from the 24 transformers that compose the GPT-2 Medium unidirectional language model [27]. Representations were built using a sliding window of 64 words as a context. We used the HuggingFace [41] implementation of this network to extract feature for these representations.

**Transformer-XL** is a set of 18 representations, each 1024-dimensional, built from each hidden state output from the 18 transformers that compose the Transformer-XL unidirectional language model [6]. Representations were built using a sliding window of 64 words as a context. We used the HuggingFace [41] implementation of this network to extract feature for these representations.

**BERT** is a set of 13 representations, each 768-dimensional, built from the encoded input and each hidden state output from the 12 transformers that compose the BERT bidirectional masked language model [10]. Representations were built using a sliding window of 64 words as a context, with the last token being the designated mask token. We used the HuggingFace [41] implementation of this network to extract feature for these representations.

**ALBERT** is a set of 13 representations, each 768-dimensional, built from the encoded input and each hidden state output from the 12 transformers that compose the ALBERT bidirectional masked language model [21]. Representations were built using a sliding window of 64 words as a context, with the last token being the designated mask token. We used the HuggingFace [41] implementation of this network to extract feature for these representations.

**Eng $\Rightarrow$ Zh** is the set of 6 encoder hidden state representations, each 512-dimensional, of the HuggingFace/Helsinki-NLP [41, 32] implementation of a Transformer-based machine translation model from English to Mandarin Chinese. Representations were built using a sliding window of 64 words as a context. The pretrained network we used can be found here: `https://huggingface.co/Helsinki-NLP/opus-mt-en-zh`.

**Eng $\Rightarrow$ De** is the set of 6 encoder hidden state representations, each 512-dimensional, of the HuggingFace/Helsinki-NLP [41, 32] implementation of a Transformer-based machine translation model from English to German. Representations were built using a sliding window of 64 words as a context. The pretrained network we used can be found here: `https://huggingface.co/Helsinki-NLP/opus-mt-en-de`.

Table 1: Stories used for stimuli for fMRI scanning

| Story | Use | Words | TRs |
|---|---|---|---|
| Alternate Ithaca Tom | Train | 2174 | 343 |
| Saving Souls | Train | 1868 | 355 |
| Avatar | Train | 1469 | 367 |
| Legacy | Train | 1893 | 400 |
| Ode to Stepfather | Train | 2675 | 404 |
| Under the Influence | Train | 1641 | 304 |
| How to Draw a Naked Man | Train | 1964 | 354 |
| Naked | Train | 3218 | 422 |
| Life Flight | Train | 2209 | 430 |
| Stage Fright | Train | 2067 | 293 |
| Till Death or Homosexuality Do Us Part | Train | 2297 | 323 |
| From Boyhood to Fatherhood | Train | 2755 | 348 |
| Sloth | Train | 2403 | 437 |
| Exorcism | Train | 2949 | 467 |
| Have You Met Him Yet | Train | 2985 | 496 |
| A Doll's House | Train | 1656 | 241 |
| The Closet that Ate Everything | Train | 1928 | 314 |
| Adventures in Saying Yes | Train | 2309 | 391 |
| Buck | Train | 1677 | 332 |
| Swimming With Astronauts | Train | 2127 | 385 |
| That Thing on My Arm | Train | 2336 | 434 |
| Eye Spy | Train | 2127 | 379 |
| It's a Box | Train | 2336 | 355 |
| Hangtime | Train | 1708 | 324 |
| Where There's Smoke | Test | 1839 | 291 |

# B    fMRI Data Details

## B.1    Stimulus details.

Training stimuli for encoding models consisted of stories from *The Moth Radio Hour*: "Alternate Ithaca Tom", "Saving Souls", "Avatar", "Legacy", "Ode to Stepfather", "Under the Influence", "How to Draw a Naked Man", "Naked", "Life Flight", "Stage Fright", "Till Death or Homosexuality Do Us Part", "From Boyhood to Fatherhood", "Sloth", "Exorcism", "Have You Met Him Yet", "A Doll's House", "The Closet that Ate Everything", "Adventures in Saying Yes", "Buck", "Swimming With Astronauts", "That Thing on My Arm", "Eye Spy", "It's a Box", and "Hangtime".

The test stimulus was a single story, also taken from *The Moth Radio Hour*: "Where There's Smoke". The amount of data in terms of words and TRs from each story is given in Table 1.

## B.2    ROI labels

Regions of interest (ROIs) were labelled in each subject using separate localizer data, as follows:

The auditory cortex (AC) was selected as a region with highly repeatably responses to ten presentations of a one-minute stimulus consisting of music, speech, and natural environmental sounds.

Visual category-selective areas were defined using a category localizer where subjects looked at a series of images while fixating on the center of the screen. Place-selective visual areas, including the retrosplenial complex (RSC), occipital place area (OPA, called TOS for trans-occipital sulcus in some subjects), and parahippocampal place area (PPA), were defined by contrasting responses to place images (e.g. buildings) with responses to object images (e.g. teapots). Face-selective visual areas, including the fusiform face area (FFA), occipital face area (OFA), and inferior frontal sulcus face patch (IFSFP) were defined by contrasting responses to face images with responses to object images. Body-selective visual areas, including the extrastriate body area (EBA) and fusiform body area (FBA) were defined by contrasting responses to body images with responses to object images.

Motor-related areas were defined using a ten-minute motor localizer in which subjects were asked to wiggle their fingers, wiggle their toes, mouth nonsense syllables (e.g. *balabalabala*), make saccades, or silently generate speech. Eye movement areas, including the frontal eye fields (FEF), intraparietal sulcus (IPS), fronto-opercular eye movement area (FO), and supplementary eye fields (SEF) were defined by contrasting the eye movement condition with rest. Hand areas, including primary somatomotor hand cortex (M1H, S1H), ventral premotor hand area (PMvh), secondary somatosensory hand area (S2H), and supplementary motor hand area (SMHA) were defined by contrasting the hand movement condition with rest. Foot areas, including primary somatomotor foot cortex (M1F, S1F), supplementary motor foot area (SMFA), and secondary somatosensory foot area (S2F) were defined by contrasting the foot movement condition with rest. Mouth movement areas, including primary somatomotor mouth cortex (M1M, S1M) and secondary somatosensory mouth cortex (S2M) were defined by contrasting the mouth movement condition with rest. Speech areas, including Broca's area (Broca) and the superior ventral premotor speech area (sPMv) were defined by contrasting the silent language generation condition with rest.

In some subjects, retinotopic areas (V1-V4, LO, V7, V3A, V3B) were defined using standard retinotopic mapping protocols.

In some subjects, the human analogue of the middle temporal visual motion area (hMT or MT) was defined using a contrast of coherently moving visual dots versus randomly flickering visual dots.

## C  fMRI Encoding Performance Similarity Matrix

A similarity matrix showing the correlations between the performance vectors for each pair of representations is given below. This can be compared to the dicriminablity matrix in Figure 4 as a ground truth metric of the the difficulty of comparing two encoding models if the representation embeddings perfectly captured all of the differences across representations. This suggests that the representation embeddings are better at capturing certain kinds of differences between representations, such as the differences across layers of the same network.

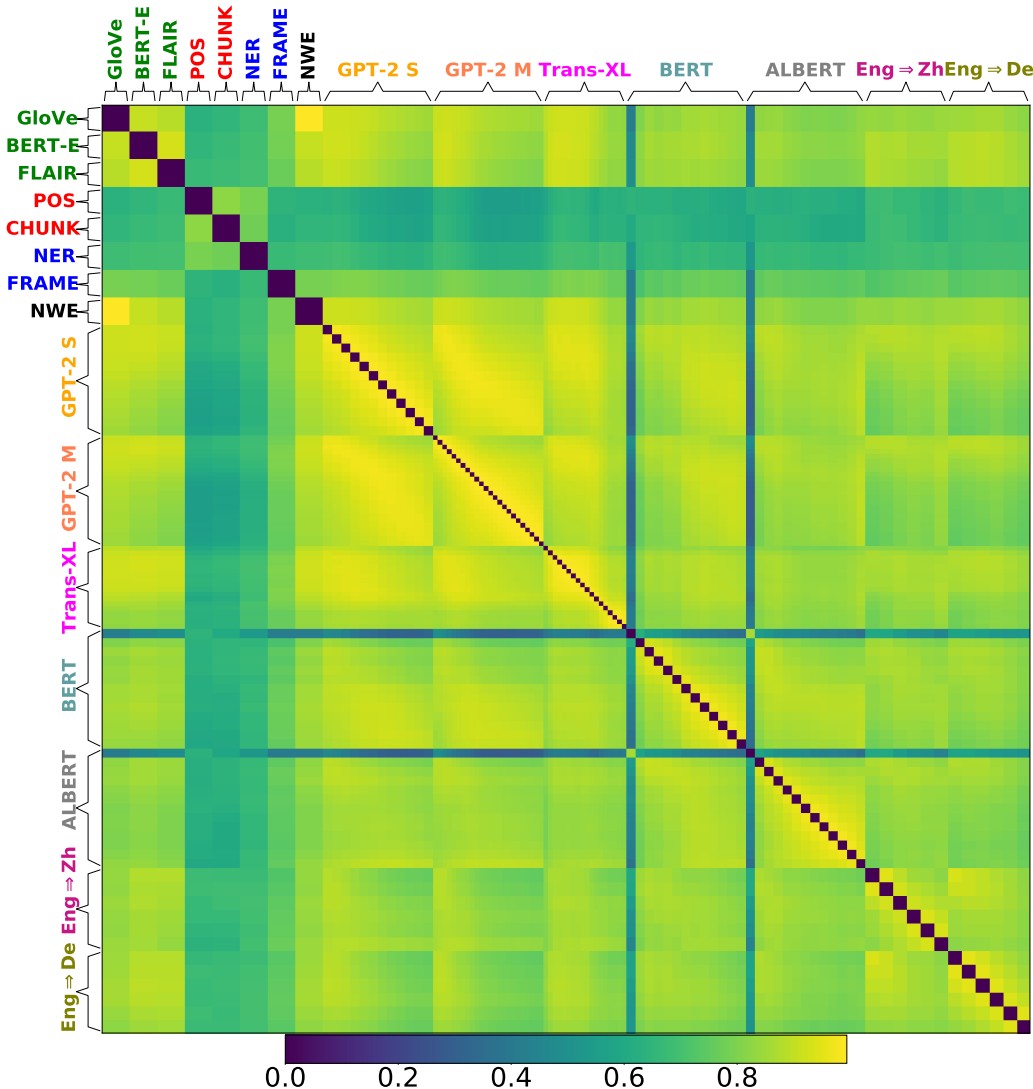

# D    Embedding Brain Voxels in the MDS Space

## D.1    Other Subjects

A 3D visualization of a single subject's brain voxels embedded into the first main MDS dimension was presented in Figure 3. Flatmaps showing the same metric the other four subjects are shown below. Scales have been adjusted on a per-subject basis to maximize visual contrast.

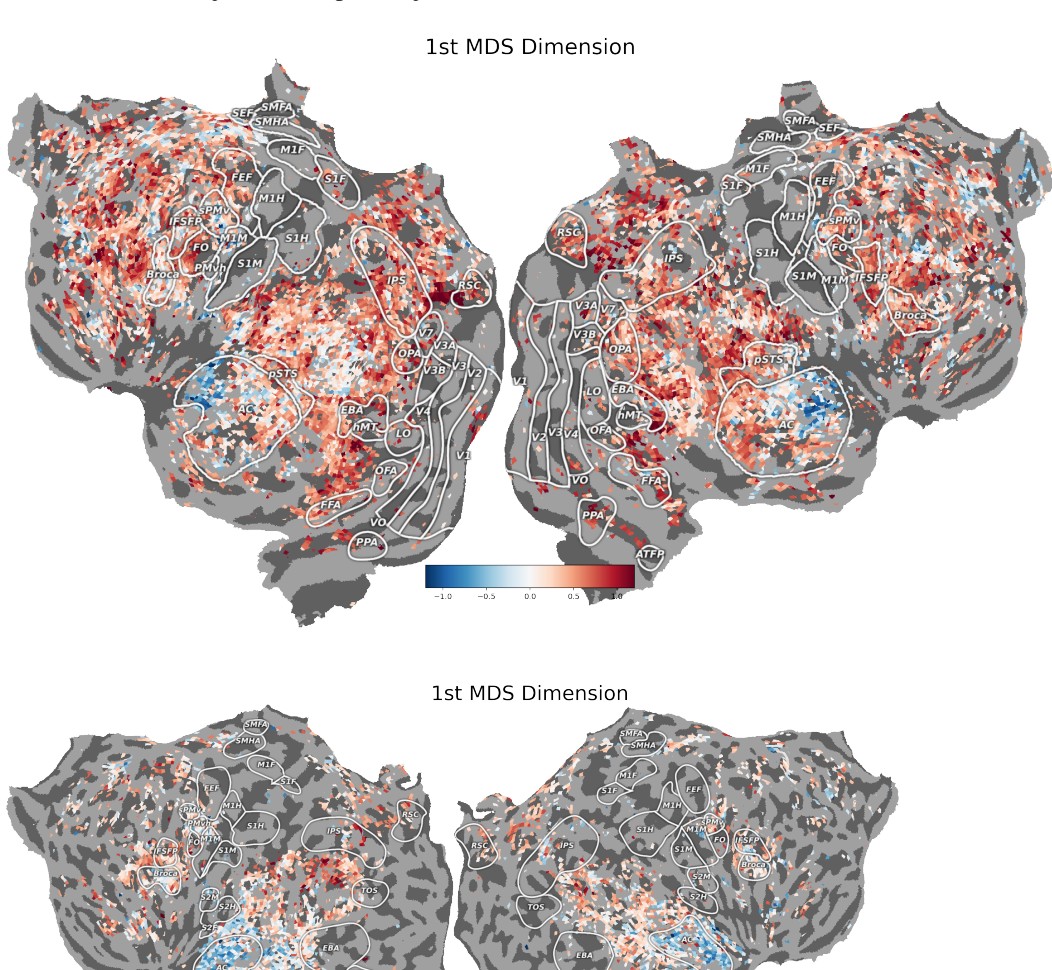

1st MDS Dimension

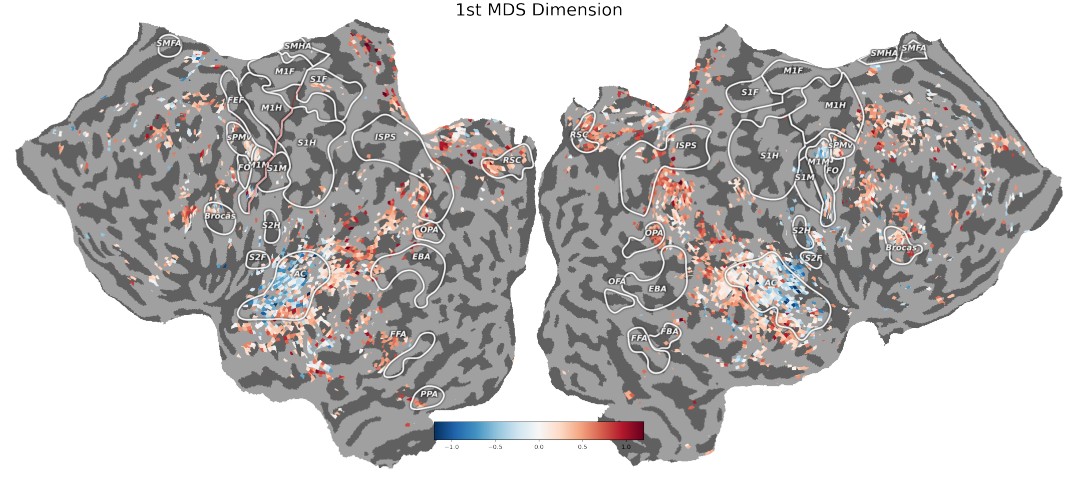

1st MDS Dimension

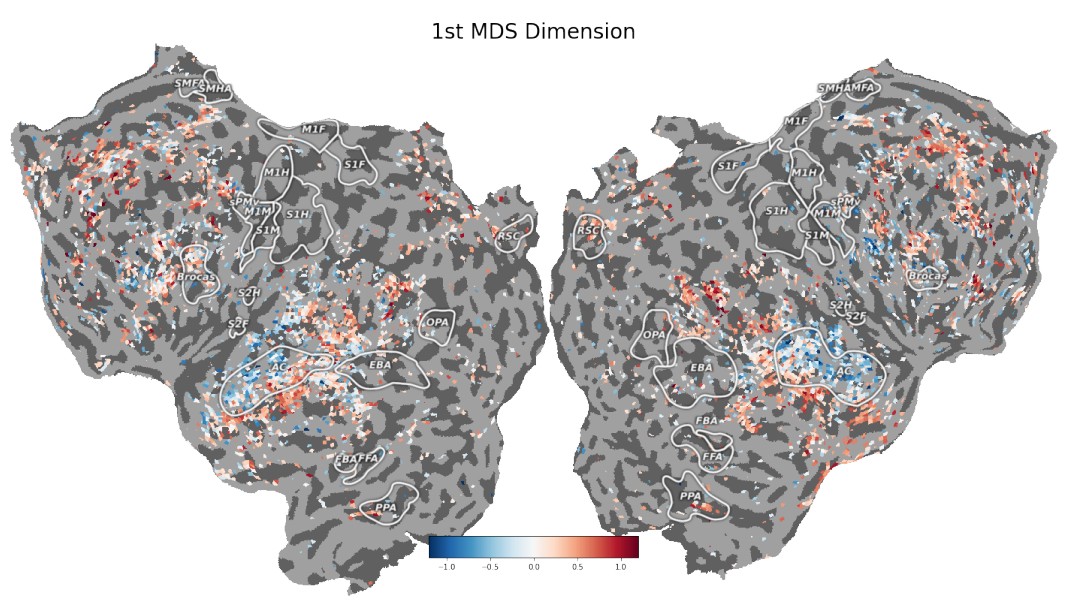

## D.2 Anatomical ROI Projection in the MDS Space

The numerical projections of the anatomical ROIs from Figure 3 are shown below. ROIs are labeled as in the Destrieux 2009 atlas [9].

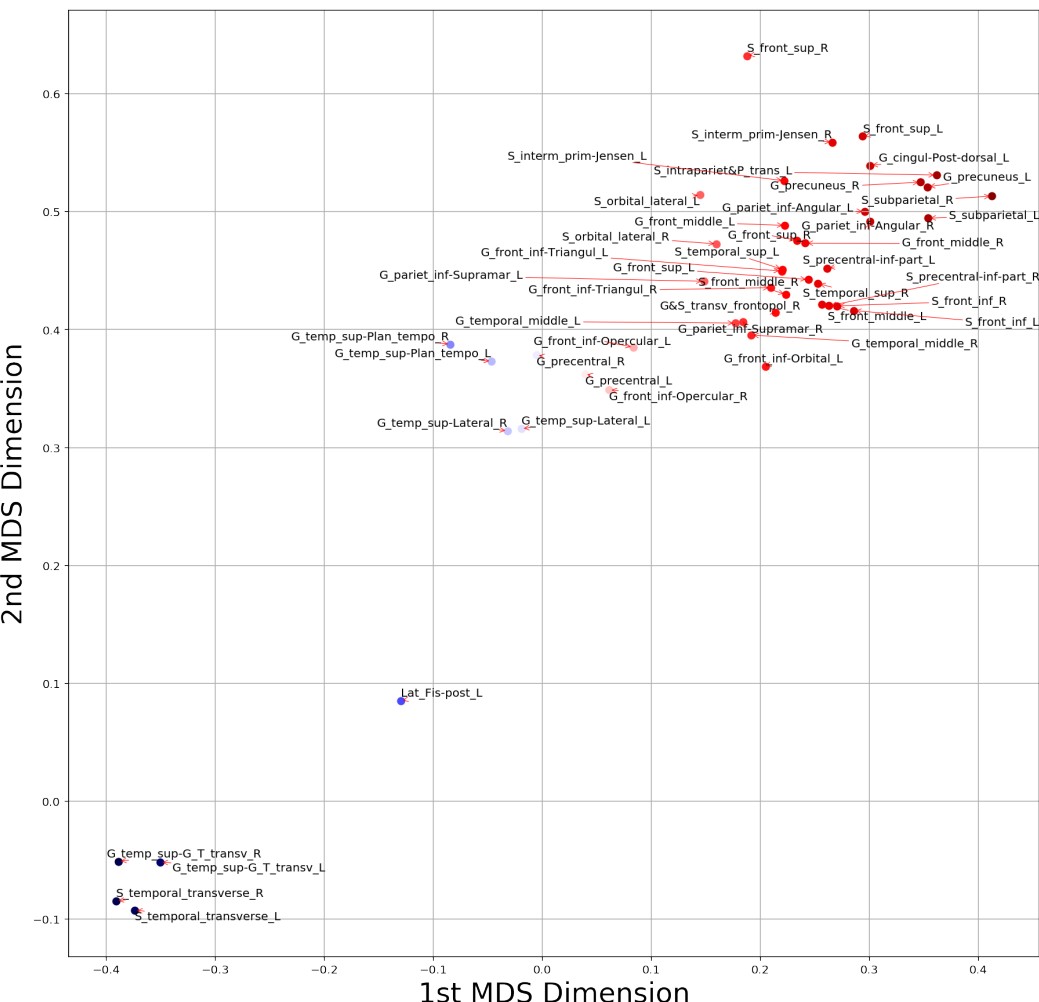

## E Experimental Setup Details

The encoder-decoder setup (before replacing decoders) was initially trained using batches of size 1024 using stochastic gradient descent using a nonstandard negative correlation coefficient loss function. We found empirically that this setup provided faster convergence and good compression into the latent spaces as it was scale-invariant to the different dimensions in the various representations. The decoders were then trained on a standard mean squared error loss after they were replaced and retrained. When decoders were retrained, *all* decoders were retrained, including decoders to the same task (e.g. $D_{t_1 \rightarrow t_1}$). This was to ensure that pairwise comparisons across decoders was performed with a consistent methodology between all pairs. Hyperparameter values were chosen via coordinate descent over a small set of values: $(10, 20, 50)$ for the latent state dimensionality, $(10^{-6}, 2 \times 10^{-5}, 10^{-4}, 2 \times 10^{-4}, 10^{-4})$ for the learning rates, for each of the encoder and decoder halves of the training, and $(256, 512, 1024)$ for batch size. Overall results did not vary significantly with latent space dimensionality. Training was done using an early stopping criterion of 1000 batches or the first batch where loss failed to improve, whichever came first. Scripts for the experimental setup have been included in the supplementary material.

# F Details on the Generation of the Pairwise Tournament Matrix W and Representation Embedding Matrix R

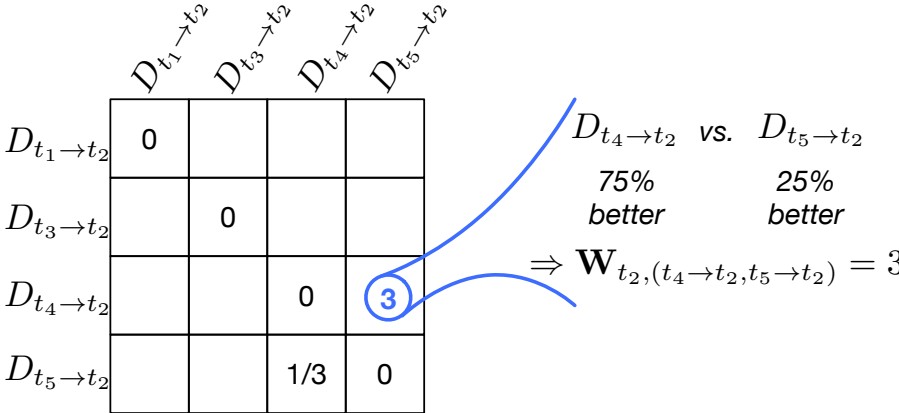

Figure 5: *Generation and Use of the Pairwise Tournament Matrix* $\mathbf{W}$*:* The decoders from a given latent space are compared in the generation of the pairwise tournament matrix for task $t_2$ $\mathbf{W_{t_2}}$. The proportion of the data for which each decoder outperforms the other is measured and compared. Diagonals of this matrix are set to 0. The eigenvector of this matrix with the highest eigenvalue is then assigned to be the row of our representation embedding matrix $\mathbf{R}$ corresponding to that matrix's encoded task. Eigenvectors and eigenvalues are computed using the differential quotient-difference algorithm. Complex components of the eigenvalue are ignored.

# G   Visual Aid Reproductions

The matrices from Figure 2 and Figure 4 are very dense. As a visual aid, these matrices have been reproduced here at a larger scale.

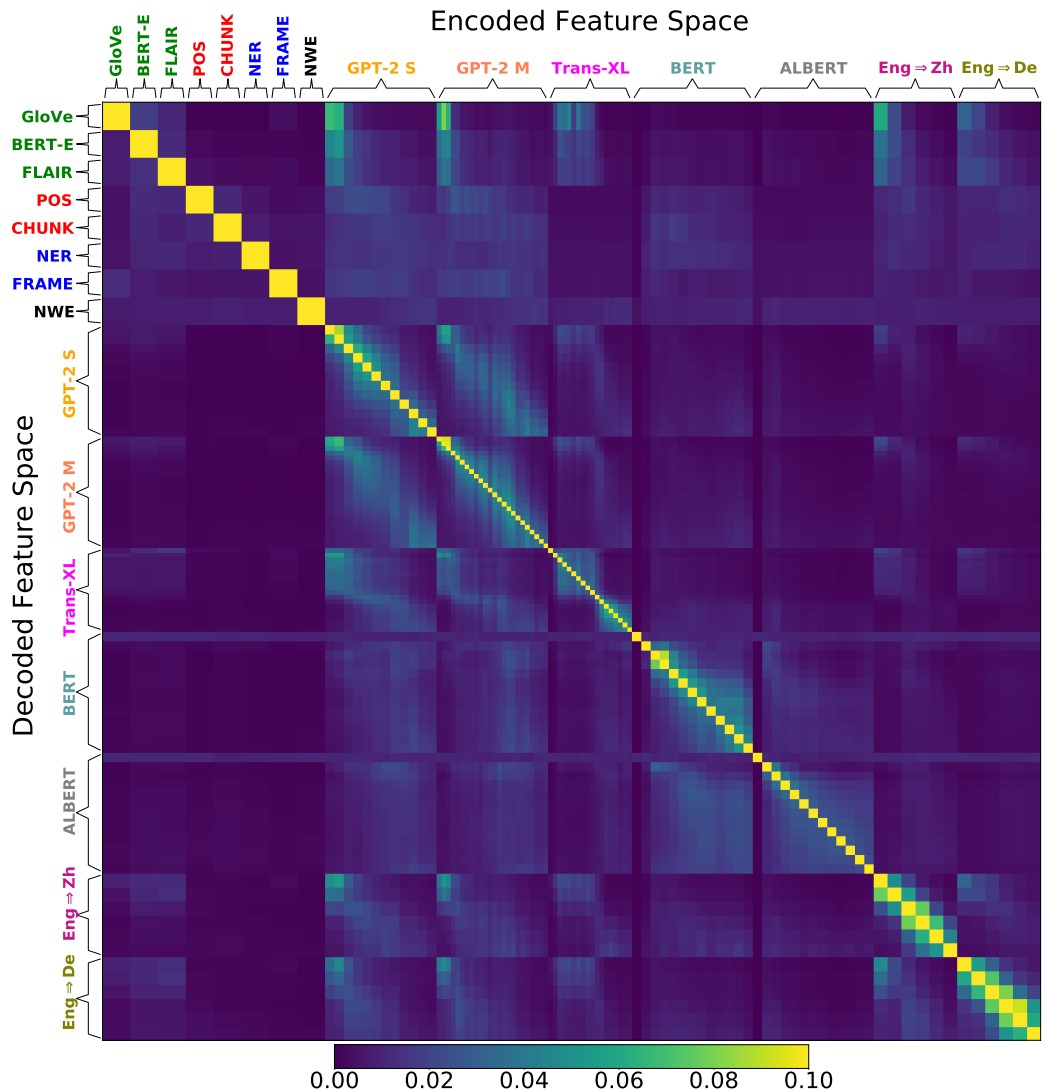

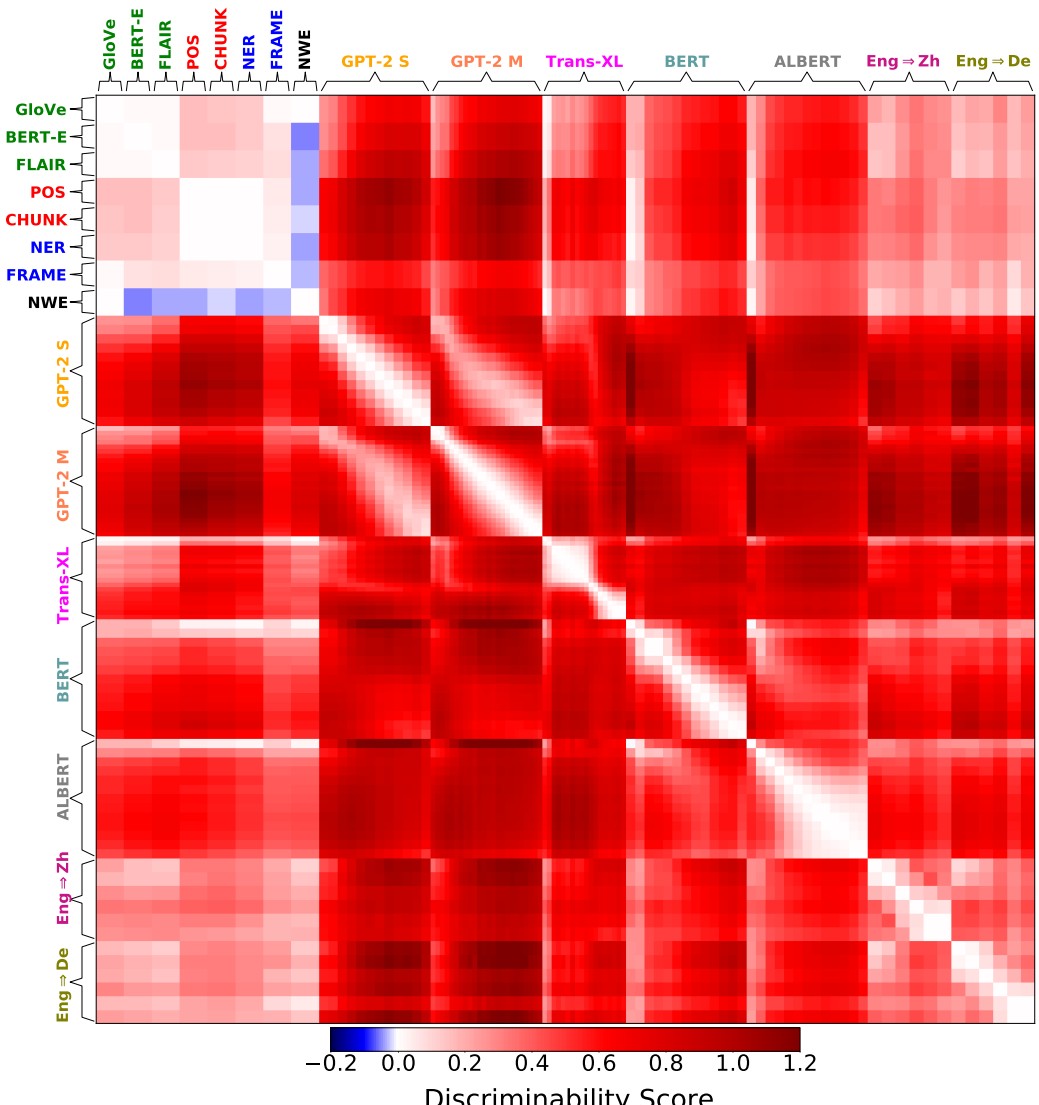