# OpenReview forum: "Low-dimensional Structure in the Space of Language Representations is Reflected in Brain Responses"
_NeurIPS.cc/2021/Conference — NeurIPS 2021 Poster_

### Official Review · Reviewer_AEZD · 2021-07-09

**Rating:** 6
**Confidence:** 4

**Summary:**

* The authors propose "representation embeddings" (vectors describing transfer performance from task i to task j) as a low-dimensional space describing the content of language representations.
* They analyze the derived representation embedding space and claim that the main principal component tracks "depth" (line 192) of linguistic processing.
* They conduct a brain encoding task and evaluate 100 models on this task. They claim that a model's brain encoding performance tracks with its behavior in this low-dimensional representation embedding space, and that this low-dimensional behavior also maps intuitively onto the spatial representation of language in the brain.

**Limitations And Societal Impact:**

See above for scientific limitations. No issues re: societal impact.

**Main Review:**

I think the paper is exceptionally well motivated among brain encoding papers, and clearly written. However, I think the results and conclusions fall short of this motivation. I think the results are marginally worth publishing as-is, but the paper could be strengthened by addressing some of the conceptual issues below.


### Quality/significance
  1. I strongly support the authors' mission to move past coarse, qualitative labels such as "syntax," "semantics," and "high/low-level" in the quest to understand both neural network and brain representations of language. But unfortunately the conclusions of the paper still seem to entirely depend on these pretheoretical notions. The MDS analysis is mainly framed as interesting because its first principal component seems to recapitulate the syntax--semantics processing trajectory (lines 183--187). The brain encoding analysis is also framed as a success because it matches up with existing notions of the spatial distribution of "higher-order" processing in the brain (line 266).As such, I think the original motivation of the authors is not fully accomplished in this paper. Ideally we would be able to use novel quantitative tools to go beyond our current conceptual framework (e.g. proving their worth in exploratory data analyses, or making predictions at a granularity / in a domain not afforded by the original qualitative distinctions).Suggestion for improvement: Explain concretely in your 'future work' section how this paper (or the next paper) can go beyond the qualitative distinctions of "syntax," "high/low-level," etc.
  2. The neuroscientific contribution is not clear to me. The authors claim to test whether the "MDS dimension could capture patterns of hierarchical processing observed in cortex," and claim that the resulting voxel map of Figure 3 answers in the affirmative. But without a more specific (quantitative / statistical?) statement of the claim and without any reference / comparison to alternative hypotheses, it's not clear what success or failure should look like for this evaluation.Suggestion for improvement: Evaluate against a quantitative baseline performance measure, or design (post-hoc?) statistical tests to make concrete some of your localization claims.
  3. I admire the creative use of higher-order "representation embeddings" here, and the mapping of its principal components back onto higher-order features of brain encoding models. But I have trouble seeing a path forward for the cognitive neuroscientific project here. This analysis setup seems to only be capable of mapping between-model representational distinctions (e.g. shallow vs deep linguistic processing) brain areas. The high-level analysis erases by design any between-stimulus distinctions; as such, it blocks us from exploring more advanced questions about fine-grained distinctions in linguistic representational content (as random examples -- how/when/where is implicature computed, how/when/where are scope ambiguities resolved, etc. etc.). Because the authors seem to be suggesting this sort of analysis as the beginning of a broader research program, I would appreciate if they could address the limits of the questions they can answer, in terms of representational content.Suggestion for improvement: Add discussion to conclusion.

### Clarity
  1. I had a very difficult time understanding both the motivation for and the mechanics of the discriminability analysis of section 4.2.4\. I don't understand what this analysis contributes beyond the more basic encoding results in Fig. 4 lower right. Why is this measure computed on pairs of representations? And why is a high discriminability evidence of "broadly captur\[ing\] the structure of the brain's language representations," over and above the basic correlation values?Suggestion for improvement: I would definitely appreciate more motivation of the particular method, which is currently very difficult to understand. (It's a second-order analysis of brain encoding models trained on second-order language representations!)
  2. Minor comments/questions
    * Line 261: What is the motivation for z-scoring these values?
    * Section 4.2.4, M equation: Why are you computing correlations here? My understanding is that both $\hat{h}$ and $\rho$ are nonrandom scalar values.

**Time Spent Reviewing:**

4

---

> ### Author Response · Authors · 2021-08-10
> **Author Response to R4**
>
> Thank you for your thoughtful review. Responses to your comments are below:
>
> > I strongly support the authors' mission to move past coarse, qualitative labels such as "syntax," "semantics," and "high/low-level" in the quest to understand both neural network and brain representations of language. But unfortunately the conclusions of the paper still seem to entirely depend on these pretheoretical notions.  Suggestion for improvement: Explain concretely in your 'future work' section how this paper (or the next paper) can go beyond the qualitative distinctions of "syntax," "high/low-level," etc.
>
> This is an excellent point. Our use of these labels in our description of these results should not be interpreted to imply that we support their general use but rather to simply give further context to the idea that the structure we have presented coincides with at least some of our natural intuition regarding the linguistic similarity of representations, much in the same way that analogies and vector projection are often used to assert that the structure of word embedding spaces is reasonable. The results in the paper can more directly be used to move beyond these coarse qualitative distinctions through the use of the representation embeddings themselves as a finer and more precise description of the characteristics of a representation. A long-term goal of the research program that this paper starts is to do exactly that: replace the imprecise notions of what it means for something to be a natural language task in NLP and cognitive neuroscience with something that is more data-driven, more standardized, and more reproducible across studies and between researchers.
>
> > The neuroscientific contribution is not clear to me. The authors claim to test whether the "MDS dimension could capture patterns of hierarchical processing observed in cortex," and claim that the resulting voxel map of Figure 3 answers in the affirmative. But without a more specific (quantitative / statistical?) statement of the claim and without any reference / comparison to alternative hypotheses, it's not clear what success or failure should look like for this evaluation. Suggestion for improvement: Evaluate against a quantitative baseline performance measure, or design (post-hoc?) statistical tests to make concrete some of your localization claims.
>
> We agree that a quantitative analysis comparing our results to known hierarchies would strengthen our claims, but we have not yet explored this direction deeply enough to offer new results. An alternative is to offer a more detailed qualitative comparison to earlier neurolinguistic findings, which we have done in our response to R2 (copied below) and would include in a final version of this paper.
>
> One point of general agreement among existing theories is that there are a set of “lower” language areas, including Wernicke’s area (which mostly overlaps with what we have labeled as auditory cortex or AC), Broca’s area in the inferior frontal lobe, and the premotor speech area sPMv, whose function is likely language-specific and related to phonology and syntax (Hickok & Poeppel, 2007; Fedorenko et al., 2012 PNAS). Across our five subjects, we find negative projections on the first PC (corresponding to earlier LM layers) in AC, but both positive and negative projections in Broca’s area and sPMv. This matches other results using narrative stories and encoding models (de Heer et al., 2017), where it was found that, of these three core language areas, AC is the best explained by lower-level features (phonemes and sound spectrum).
>
> Outside of the core language areas, the literature is more divided on representational hierarchy. It is broadly agreed that these other areas constitute the “semantic system” (Binder et al., 2009), in which language meaning is derived and represented. This fits with our data, which show that most of these regions have positive projections on the first PC, corresponding to later LM layers. Yet some also claim specific high-level roles for areas like the angular gyrus, which is thought to be a site where information from multiple words or senses is combined (Price et al., 2015 J. Neurosci.). Interestingly, our data actually assign negative projections (earlier LM layers) to the angular gyrus, unlike nearly all of the surrounding cortex. This could reflect a more “core” role of angular gyrus in language processing, similar to Wernicke’s or Broca’s areas.
>
> In future work, we also plan to quantitatively compare our data to other neuroanatomical (e.g. cortical myelination) and functional (e.g. primary connectivity gradient, Margulies et al., 2016) measures of hierarchy.
>
> > What is the overall takeaway from the rankings of the 100 representations in terms of their power as predictors in neural encoding models?
>
> Here we believe you are referring to the bar plot showing the average prediction performance of encoding models trained with each of the representational spaces (Figure 4, bottom right panel). We did not intend to highlight this as a major result. Indeed, it was included largely to show that our results conform to earlier findings (e.g. Toneva et al., 2019; Schrimpf et al., 2020). Our focus in these analyses was more on the pattern of prediction performance across the brain (what is being discriminated), rather than the average value. This analysis shows that these patterns, for most pairs of representations, are distinct across cortex, and that the distinctions are sufficiently related to the task affinities that we can “decode” which feature space was used in a model from the pattern of prediction performance.
>
>
> >  This analysis setup seems to only be capable of mapping between-model representational distinctions (e.g. shallow vs deep linguistic processing) brain areas. The high-level analysis erases by design any between-stimulus distinctions; as such, it blocks us from exploring more advanced questions about fine-grained distinctions in linguistic representational content (as random examples -- how/when/where is implicature computed, how/when/where are scope ambiguities resolved, etc. etc.). Because the authors seem to be suggesting this sort of analysis as the beginning of a broader research program, I would appreciate if they could address the limits of the questions they can answer, in terms of representational content. Suggestion for improvement: Add discussion to conclusion.
>
> We disagree with the particulars of this assessment. The scope of what a representation can be in our setup is extremely broad, essentially any function from our linguistic stimulus. For instance, to use an example that you specify, one could define a linguistic representation specifically targeted to scope ambiguities such as the pre-softmaxed logits of a neural network trained to identify or classify the resolution of scope ambiguities. This raises the limitation that in order to study a distinction such as this, one would first have to curate the necessary representation to isolate that particular distinction
>
> > I had a very difficult time understanding both the motivation for and the mechanics of the discriminability analysis of section 4.2.4. I don't understand what this analysis contributes beyond the more basic encoding results in Fig. 4 lower right.
>
> We apologize for the lack of clarity on this particular analysis. We have added here our discussion with R1 on the subject. We intend to resolve any issues with clarity in a camera-ready version.
> The discriminability score has a somewhat verbose mathematical definition but this is simply to ensure fairness in how we test the representation embeddings. The high-level idea here is that if the representation embeddings actually reflect the structure of how the brain represents linguistic information, then we should be able to “match” each representation embedding to its corresponding performance vector, which describes which regions contain the information in a given representation. We structure this as a leave-two-out experiment, where we train a learner using 98 (representation embedding, performance vector) pairs and then use this learner to try and predict which of the remaining two representation embeddings match with the remaining two performance vectors. If the correlation between the predicted performance vector for representation A matches the ground truth performance vector for representation A better than the ground truth performance vector for representation B (and similarly B matches B better than it matches A), then we have succeeded in correctly discriminating a performance vector from its corresponding representation embedding.
>
> The negative scores represent below-chance performance at discriminating between that pair of representations. This could happen by chance: if two representations are truly not discriminable in the brain, then the discrimination scores should be randomly distributed around zero. But it could also happen if the task affinities are very different from the brain representations. We believe the negative scores for GloVe NWE are caused by the latter. This representation is highly dissimilar from the others in the type of information that it contains, hence its far-removed position in the MDS space. However, the temporal imprecision of the BOLD responses recorded by fMRI (which is on the order of 1 second, or 2-5 words) makes it so that the GloVe NWE and normal GloVe embeddings are functionally very similar for predicting brain data (note that the average prediction performance for NWE and normal GloVe is almost identical in the lower right panel of Figure 4). Because our discrimination test uses the affinities to predict the pattern of fMRI encoding performance across the brain, this disconnect—NWE is different from everything in the affinity analysis, but very similar in the fMRI analysis—is likely causing the negative scores. We will clarify this in the final version of the paper.

---

> > ### Author Response · Authors · 2021-08-10
> > **Author Response to R4 (2)**
> >
> >
> > > Line 261: What is the motivation for z-scoring these values?
> >
> > Z-scoring these values allows us to bring these performance vectors into the same relative space as the representation embeddings prior to projection into the MDS space.
> >
> > > Section 4.2.4, M equation: Why are you computing correlations here? My understanding is that both h^ and ρ are nonrandom scalar values.
> >
> > They, h^ and \rho, are voxelwise *vectors* with a dimensionality equal to the number of voxels. Correlations are used to determine the similarity of the voxelwise performance vector predicted from the representation embedding against the corresponding ground truth performance vector.

---

### Official Review · Reviewer_aJCH · 2021-07-10

**Rating:** 6
**Confidence:** 4

**Summary:**

The submission proposes to study similarities between language representations in different language models. First, a similarity measure between language representations is derived. Then, the matrix of similarities is reduced to two dimensions using MDS. The authors then discuss qualitatively this low-dimensional space of language representations. Then, linear encoding models of fMRI recordings are used to project the first MDS dimension on the cortical surface of a subject listening to narratives. Finally, the matrix of similarities is used to predict the performances of linear encoding models.


**Limitations And Societal Impact:**

Yes

**Main Review:**

*Originality:* Comparing representations from intermediate layers of deep neural networks is an interesting topic and has received a lot of interest in the recent years, as adequately discussed in the introduction. This submission proposes a new method to compare representations, using a procedure based on bottle-necked encoder-decoder across representations, creating tournament matrices from pairwise comparisons of representations, and stacking the tournament matrices first eigenvectors to create a similarity matrix. This procedure seems partially novel and partially inspired from Zamir et al [38], although it is not clear how this work exactly relates to previous contributions. In particular, there is no quantitative nor qualitative comparison with what would be obtained with CCA or CKA.

*Quality:* The submission is technically sound, although the motivation of each step of the procedure could be better explained. The results look reasonable and robust, and adequate care about generalization and overfitting is mentioned. The authors note the limitation of centering their similarity measure around the GloVe word embedding. The shared code and data is appreciated, although not all code files reach a reasonable level of documentation. Importantly however, the authors do not compare their procedure to other similarity of representations methods .

*Clarity:* The paper is clearly written and easy to follow, except a few convoluted sentences (e.g. L36, L74). However, the paper could better inform the reader about what parts of the method to build the similarity matrix is novel. The discriminability scores in section 4.2.4 are not enough motivated and explained. In particular, it is not clear why "high discriminability scores (red) [indicates] that the representation embeddings broadly capture the structure of the brain’s language representations".

*Significance:* It is hard to judge the significance of the method section (section 3), since no comparison with previous work is proposed. However, using MDS on the similarity matrix and the corresponding discussion around Figure 2 is interesting. The projection over the cortical surface is not surprising, as the first MDS dimension is mainly a measure of the DNN layer, and similar gradients have been described over encoding models built from DNN layers [e.g. Kell et al, Neuron 2018].

*remark:* NLP is not defined.

**Time Spent Reviewing:**

2.5

---

> ### Author Response · Authors · 2021-08-10
> **Author Response to R3**
>
> Thank you for your thoughtful review. Comments to your concerns are below:
>
> > In particular, there is no quantitative nor qualitative comparison with what would be obtained with CCA or CKA. / Importantly however, the authors do not compare their procedure to other similarity of representations methods .
>
> We assume you are referencing the work by Kornblith et al. in their paper on the similarity of neural network representations. We chose this representation embedding space over alternative ways of representing representations because we believed it to be better suited to what we intended to accomplish in comparing task-based representations, as it has been used in past papers by Zamir et al. and others to great success. The goal of this portion of the paper was not to compare or analyze all possible ways of computing representational similarity, and we believe a focus on that would represent a deviation from the main objectives of the paper.
>
> > The shared code and data is appreciated, although not all code files reach a reasonable level of documentation.
>
> We will improve on this in a final release with a camera-ready version.
>
> > The discriminability scores in section 4.2.4 are not enough motivated and explained. In particular, it is not clear why "high discriminability scores (red) [indicates] that the representation embeddings broadly capture the structure of the brain’s language representations".
>
> The discriminability score has a somewhat verbose mathematical definition but this is simply to ensure fairness in how we test the representation embeddings. The high-level idea here is that if the representation embeddings actually reflect the structure of how the brain represents linguistic information, then we should be able to “match” each representation embedding to its corresponding performance vector, which describes which regions contain the information in a given representation. We structure this as a leave-two-out experiment, where we train a learner using 98 (representation embedding, performance vector) pairs and then use this learner to try and predict which of the remaining two representation embeddings match with the remaining two performance vectors. If the correlation between the predicted performance vector for representation A matches the ground truth performance vector for representation A better than the ground truth performance vector for representation B (and similarly B matches B better than it matches A), then we have succeeded in correctly discriminating a performance vector from its corresponding representation embedding.
>
> The negative scores represent below-chance performance at discriminating between that pair of representations. This could happen by chance: if two representations are truly not discriminable in the brain, then the discrimination scores should be randomly distributed around zero. But it could also happen if the task affinities are very different from the brain representations. We believe the negative scores for GloVe NWE are caused by the latter. This representation is highly dissimilar from the others in the type of information that it contains, hence its far-removed position in the MDS space. However, the temporal imprecision of the BOLD responses recorded by fMRI (which is on the order of 1 second, or 2-5 words) makes it so that the GloVe NWE and normal GloVe embeddings are functionally very similar for predicting brain data (note that the average prediction performance for NWE and normal GloVe is almost identical in the lower right panel of Figure 4). Because our discrimination test uses the affinities to predict the pattern of fMRI encoding performance across the brain, this disconnect—NWE is different from everything in the affinity analysis, but very similar in the fMRI analysis—is likely causing the negative scores. We will clarify this in the final version of the paper.
>
> > NLP is not defined.
>
> Here we use NLP to mean natural language processing. We will define it upon first use in a camera-ready submission.

---

### Official Review · Reviewer_KDff · 2021-07-15

**Rating:** 7
**Confidence:** 4

**Summary:**

Using a slightly modified (but relatively straightforward version) of an analysis that’s recently been called ‘neural taskonomy’, this work attempts to use the relationships (read: transfer learning affinity) among different natural language representations to assess the organization of linguistic representation in human fMRI data via regularized linear encoding models.  The authors first contribute a modified version of a previously used encoder-decoder framework to create a 'task' (linguistic representation) affinity matrix that demonstrates conspicuous, interpretable structure when reduced via MDS to fewer projections. They then use the structure discernible in this matrix to ascertain whether similar structure (specifically, a hierarchy of representation from word embeddings to later, deep transformer layers) is evident in the brain.

**Ethical Concerns:**

The authors adequately addressed ethical concerns.

**Limitations And Societal Impact:**

Yes -- though perhaps some thoughts on the limitations of purely data-driven approaches to computational neuroscience would have been apropos.

**Main Review:**

The ‘taskonomy’ portion of this paper (in that it closely mirrors the work of Zamir & Colleagues) is mostly clear and well-constructed; the selection of models seems comprehensive enough (spanning relatively simple word embeddings to more complex transformers), and the range of linguistic tasks covered seems (at first glance) sufficient to recapitulate the diversity of semantic and syntactical structures that should presumably define the neural code. Some minor points of clarification I think could benefit this section are:
- Clarifying what exactly is meant by the ‘mean squared error’ on some % of the data in the generation of the pairwise tournament matrix (135-139). How is the data being divided? What exactly is the data in this case? A particular set of sentences in the corpus being tested? This section – though perhaps routine in principle – left me utterly complexed.
-Reformulating the sentence starting at line 161 (“We also included the GloVe 162 embedding for the next word in the sequence, which constitutes a low-dimensional representation the ideal output of a language model”) to both remove the typo (assuming it’s missing a colon) and to clarify why a low-dimensional representation is the ideal output of a language model. This is not immediately clear to me a priori, and I might even be inclined to say the opposite, depending on the functional context in which the representation is leveraged.
-Is the vertical (top to bottom) dimension in the MDS interpretable? The EFA suggests the second most significant factor still accounts for 10% of the variance. Does this 10% come with interpretability? If not, is it really the case that the low-dimensional structure among the natural language representations is interpretable in only one direction?
-What is so different about the GLOVE Next Word embeddings? Why their consignment to the periphery of the MDS?

All in all, I find this section particularly compelling, and am intrigued by the structure the authors seemed to have uncovered – though an attempt at interpreting other dimensions after the 1st could make it just a bit more satisfying.

The ‘neural’ portion of this paper is the less compelling of the two – not necessarily due to methodological flaws, but to a lingering sense that some of the author’s central claims (including, most importantly, the recapitulation of the taskonomic representational hierarchy across the brain) are not so neatly supported by the data. Take for example the BrainView in Figure 3. The authors make a significant point of underscoring a hierarchy of representation (from word embeddings to late transformer layers), but most of the areas activated by the ‘later’ (presumably more complex) representations are early perceptual areas (including most conspicuously, early visual cortex). Anterior inferotemporal cortex (the main site of object recognition, and presumably the location where vision interfaces with language) is dominated by ‘earlier’ (presumably less complex) representations. I can’t quite make sense of this in a narrative about representational hierarchy – but this brings me to a second point: the authors do not make clear what we should be looking for. Are there known representational hierarchies in the neuroimaging of language? If so, how are they characterized and what are their stereotypical motifs? Obviously, the specific loci of syntactic and semantic structures in the brain remain a matter of significant (even hot) debate – but some reference point here (from Caramazza, Federenko or other similar researchers) could help to situate and better direct the reader’s attention to key data points of high interpretive value.

Similarly, the lack of an overall gestalt in Figure 4 (at least a gestalt comparable to that provided by the MDS plot in Figure 2) gives the impression that the authors relied a bit too heavily on a data-driven approach without a clear sense of the theoretical insights to be gained from their otherwise powerful methodologies. What does it mean that the GPT models were overall most predictive of neural activity? Is there discernible structure in the MDS computed on the discriminability score matrix (perhaps structure comparable to that found for transfer affinities)? For that matter, what do all of these answers look like in parts of the brain localized or functionally segmented with a language-specific task? Without answers to these questions, I find it difficult to say what I’ve learned from the ‘neural’ half of this ‘neural taskonomy’ analysis.

I think this paper has significant potential, and would be happy to add to my score if the authors can more clearly address the following two points in their rebuttals:
- Is there a distinct linguistic hierarchy in (either the whole or a functionally localized linguistic portion of) the brain? If yes, how is it defined, where exactly is it located and does the hierarchy unveiled by your task affinities match this hierarchy?
- What is the overall takeaway from the rankings of the 100 representations in terms of their power as predictors in neural encoding models?


**Time Spent Reviewing:**

4

---

> ### Author Response · Authors · 2021-08-10
> **Author Response to R2**
>
> Thank you for your thoughtful review. We have answered your questions in order below:
>
> > Clarifying what exactly is meant by the ‘mean squared error’ on some % of the data in the generation of the pairwise tournament matrix (135-139). How is the data being divided? What exactly is the data in this case? A particular set of sentences in the corpus being tested?
>
> Yes, this corresponds to a particular set of held out sentences/contexts (represented by the corresponding representation) from an entirely separate test story of 1839 words. We will include this detail in a camera-ready version.
>
> > Reformulating the sentence starting at line 161 (“We also included the GloVe 162 embedding for the next word in the sequence, which constitutes a low-dimensional representation the ideal output of a language model”) to both remove the typo (assuming it’s missing a colon) and to clarify why a low-dimensional representation is the ideal output of a language model. This is not immediately clear to me a priori, and I might even be inclined to say the opposite, depending on the functional context in which the representation is leveraged.
>
> The sentence quoted should actually read “...which constitutes a low-dimensional representation OF the ideal output of a language model". The ideal output is the next word, the low-dimensional representation is the glove embedding. We will fix this error.
>
> > Is the vertical (top to bottom) dimension in the MDS interpretable? The EFA suggests the second most significant factor still accounts for 10% of the variance. Does this 10% come with interpretability? If not, is it really the case that the low-dimensional structure among the natural language representations is interpretable in only one direction?
>
> We are not sure if it is easily interpretable, and try not to make such claims in the paper. Instead, it may be more useful to consider the space as a whole, which mostly resides within a triangle that has vertices at the word embeddings, the interpretable spaces (POS, Chunking, Framing, NER), and the next-word embedding. The language models span this triangular space, starting at the word embeddings, arcing toward the interpretable spaces, then arcing toward the next-word embedding. These arcs proceed almost uniformly along the first dimension, from negative (lower LM layers) to positive (higher LM layers). However, on the second dimension the values actually rise and then fall, with the highest values being assigned to the middle layers (e.g. layer 5 of 12 in GPT-2 small). The MDS space suggests that those middle layers are the most similar to the interpretable, human-defined representations (POS, NER, etc.), while the earlier and later layers are less similar. If the human-defined representations are the most abstracted from the input word sequence (i.e. because they are the lowest-dimensional), then this might suggest that the second dimension is capturing something akin to level of abstraction. We will include discussion of this issue in the final version of this paper. We also welcome any suggestions for probes or metrics that can be applied to these representations!
>
>
>
> > What is so different about the GLOVE Next Word embeddings? Why their consignment to the periphery of the MDS?
>
> It is very reasonable to believe the next word embedding (NWE) is and should be far from the other representations. Relative to the non-language-model representations, NWE contains information that is fundamentally different, since it is computed from different words. But it also makes sense that NWE is rather far from the LM output representations. This is because the language modelling task is fundamentally uncomputable. If a language model is asked to finish the sentence “The car is ____.”, its hidden state representation must represent the distribution over all possible completions. It can be black/white/big/small, and none of these possibilities are more wrong than the others based on the information available to the language model. So the LM representations are going to include information about that uncertainty, whereas the NWE representation will not, as it has special oracle access to this uncomputable ground truth.
>
> > Are there known representational hierarchies in the neuroimaging of language? If so, how are they characterized and what are their stereotypical motifs? Obviously, the specific loci of syntactic and semantic structures in the brain remain a matter of significant (even hot) debate – but some reference point here (from Caramazza, Federenko or other similar researchers) could help to situate and better direct the reader’s attention to key data points of high interpretive value. / Is there a distinct linguistic hierarchy in (either the whole or a functionally localized linguistic portion of) the brain? If yes, how is it defined, where exactly is it located and does the hierarchy unveiled by your task affinities match this hierarchy?
>
> This is an excellent set of questions, and one that we agree we did not engage with sufficiently in the initial submission. As the reviewer points out, the literature has not settled on a single hierarchical map of language processing in human cortex. Yet an effort to identify organizational features that appear in our results as well as proposed hierarchies would help position our results.
>
> One point of general agreement among existing theories is that there are a set of “lower” language areas, including Wernicke’s area (which mostly overlaps with what we have labeled as auditory cortex or AC), Broca’s area in the inferior frontal lobe, and the premotor speech area sPMv, whose function is likely language-specific and related to phonology and syntax (Hickok & Poeppel, 2007; Fedorenko et al., 2012 PNAS). Across our five subjects, we find negative projections on the first PC (corresponding to earlier LM layers) in AC, but both positive and negative projections in Broca’s area and sPMv. This matches other results using narrative stories and encoding models (de Heer et al., 2017), where it was found that, of these three core language areas, AC is the best explained by lower-level features (phonemes and sound spectrum).
>
> Outside of the core language areas, the literature is more divided on representational hierarchy. It is broadly agreed that these other areas constitute the “semantic system” (Binder et al., 2009), in which language meaning is derived and represented. This fits with our data, which show that most of these regions have positive projections on the first PC, corresponding to later LM layers. Yet some also claim specific high-level roles for areas like the angular gyrus, which is thought to be a site where information from multiple words or senses is combined (Price et al., 2015 J. Neurosci.). Interestingly, our data actually assign negative projections (earlier LM layers) to the angular gyrus, unlike nearly all of the surrounding cortex. This could reflect a more “core” role of angular gyrus in language processing, similar to Wernicke’s or Broca’s areas.
>
> In future work, we also plan to quantitatively compare our data to other neuroanatomical (e.g. cortical myelination) and functional (e.g. primary connectivity gradient, Margulies et al., 2016) measures of hierarchy.
>
> > What is the overall takeaway from the rankings of the 100 representations in terms of their power as predictors in neural encoding models?
>
> Here we believe you are referring to the bar plot showing the average prediction performance of encoding models trained with each of the representational spaces (Figure 4, bottom right panel). We did not intend to highlight this as a major result. Indeed, it was included largely to show that our results conform to earlier findings (e.g. Toneva et al., 2019; Schrimpf et al., 2020). Our focus in these analyses was more on the pattern of prediction performance across the brain (what is being discriminated), rather than the average value. This analysis shows that these patterns, for most pairs of representations, are distinct across cortex, and that the distinctions are sufficiently related to the task affinities that we can “decode” which feature space was used in a model from the pattern of prediction performance.

---

> > ### Comment · Reviewer_KDff · 2021-08-13
> > **Discussion on First Response**
> >
> > Thanks to the authors for their detailed response. I am largely satisfied with the clarifications on the modeling, and appreciate the added details. I am still curious about the quadratic peak along the 2nd axis of the MDS, but appreciate these kinds of trends are often difficult to interpret.
> >
> > As for the neural data, the added information about proposed hierarchy is a KEY addition that is starting to change my mind, but is still hard to process in its current form. As a central claim of the abstract (and the paper writ large) I would like to see a figure (nothing formal, just a sketch) that could potentially replace the score plots in figure 4 (which strike me as largely uninformative, and potentially better situated in the appendix since they're meant to show concordance with previous results) and makes clear precisely how the 1st dimension of the MDS recapitulates the proposed information processing hierarchy across language areas in the brain. One option for this figure would be a simple lineplot with the value of the MDS along the y axis, and the proposed hierarchy (Wernicke's -> Broca's -> ?) along the x axis, ordered by their hypothesized rank in the hierarchy. Points at each of these discrete units could have errorbars that represent the average value of the MDS summarized across all voxels in the ROI. Individual lines (separated by color) could be plotted per subject. A figure of this variety would be entirely sufficient for me to upvote this paper to full accept.
> >
> > Less importantly, I still find myself a bit perplexed by some of the activations in the fMRI. (Why does early visual cortex correspond to later layers in the deep language models?) But, I understand that brain data is noisy, so if the above figure seems reasonable, I'm willing to look past these other elements.

---

> > > ### Author Response · Authors · 2021-08-18
> > > **New figure showing MDS projections of different brain areas**
> > >
> > > Thank you for the suggestions! Building on these ideas, we created a new figure that summarizes the MDS projections of 47 anatomical ROIs (25 in the left hemisphere, 22 in the right). To preserve anonymity, we have posted it here: https://imgur.com/a/HgW7Ewq Please note that this is still draft quality, and will be refined for the final version of the paper.
> > >
> > > The right side of this figure shows the projection of each anatomical ROI onto the first and second MDS dimensions. Projections are averaged across high-signal voxels in each ROI (voxel selection is performed as described elsewhere in the paper) and across the five subjects. We only included ROIs where, in a majority of subjects (i.e. at least 3 out of 5), at least X% of voxels were high-signal, where X% is the total percentage of high-signal voxels across that subject’s brain. This cut the original 76 ROIs down to 25 in the left hemisphere and 22 in the right. Anatomical ROIs were defined automatically in each subject using Freesurfer with the Destrieux 2009 atlas (“aparc.a2009s”; Destrieux et al., 2010). The marker for each area is colored according to its projection on the first MDS dimension (identical to its location along the x-axis). The left side of the figure shows the flattened cortex for one subject where each ROI is assigned the same color as its marker on the right.
> > >
> > > What the MDS projections show is that cortical areas largely form a linear progression from bottom left (low on both dimensions) to top right (high on both dimensions). This progression contains the classical hierarchy discussed in our earlier reply: early auditory processing (the superior temporal areas) appears at the bottom, core language areas (Broca’s area, extended portions of “Wernicke’s” area, sPMv) appear towards the middle, and high-level semantic areas (precuneus, angular gyrus) appear at the top. We believe this strongly supports our earlier assertion that the taskonomy MDS has uncovered something akin to the brain’s language representation hierarchy.
> > >
> > > However, we also note some issues with this interpretation. Averaging values across ROIs and across subjects strongly smooths the results. Individual subjects—for whom we have enough data to estimate the values in each voxel with relatively high fidelity—show that there can be substantial variation within each of these anatomical ROIs. Differences in data quality between subjects also limits what we can see in the average. For example, the visual cortex areas you noted earlier are removed here because they are not consistently responsive to language across subjects. It is also clear from the individual subject maps (Figure 3 in the original submission) that regions like the prefrontal cortex actually contain a more detailed map, with low-MDS-1 areas interspersed with high-MDS-1 areas. These details are lost when combining data across subjects using crude anatomical ROIs (though note that this approach is common in the field). Finding better ways to parcellate cortex consistently across subjects and understand how language is processed across its surface will be important in future work.

---

> > > > ### Comment · Reviewer_KDff · 2021-08-31
> > > > **Continuing Discussion**
> > > >
> > > > Thanks to the authors for their detailed response. All complexities of averaging across fMRI data aside, the figure added by the authors seems to me a promising first start, and I encourage the authors to include it in the main manuscript. I still think a categorical x axis, with cortical areas organized according to the proposed hierarchy would be informative, but I think what the authors have provided is sufficient as a demonstration they're intent on making the larger point about hierarchy more salient. I also still think some explanation of the MDS projections elsewhere (e.g. in early visual cortex) is in order, but am for the moment satisfied enough to increase my score and vote for acceptance.

---

### Official Review · Reviewer_u2XG · 2021-07-17

**Rating:** 7
**Confidence:** 3

**Summary:**

This paper is about two things: understanding learned NLP representations by finding and understanding similarities between *different* representations; and comparing these representations to fMRI data. The paper takes a new approach to both problems, and finds a dimension in the space of language representations that seems to correspond to something like "semantic depth of analysis." Furthermore, the comparison with fMRI data shows that many representations are correlated with brain activations.

**Limitations And Societal Impact:**

Yes, though see above.

**Main Review:**

This paper is very interesting methodologically, in the sense that it brings novel methods to an important problem, and also in the sense that these methods are highly worthy of further examination. I don't feel that I learned much about NLP representations or about brain representations, but I do feel like the approach taken here is going to be an important part of answering these questions in the future.

The paper is very clear, but there were still some parts that I had difficulty with. First of all, the methods in 3.3. It seems complicated: if the goal is to get a vector, for each type of representation, that encodes in some sense how similar another representation is, then why go to the trouble of the "tournament"? Why not just, for each element i, directly use the MSE for decoding the target representation using the representation i? And, for that matter, even in the tournament version, why not take into account the actual values of the MSEs rather than just the percentage for which the one was higher than the other? Given that there is a simpler idea at hand, I would guess that the tournament approach must be necessary. Otherwise the paper would have presumably taken a simpler approach. But I do not understand why. This leaves open the possibility that this approach was chosen empirically, which wouldn't be a good thing here. More explanation of these methods and whether and why they *must* be the way they are, is necessary.

More information would be welcome about what the "temporal transformation" is in 4.2.2.

In 4.2.3, the methodology is sufficiently complicated that I have a hard time following what exactly \rho is. More importantly, I have a hard time understanding why the \rho values should be sufficiently comparable to the representation embeddings that it would make any sense to project them onto the first basis vector of the MDS. I have a very strong feeling that it makes sense,but this part of the paper just goes too fast for me to understand exactly with these p vectors are.

Similarly, I have a hard time getting the intuitions behind the method in 4.2.4, and, as such, I don't feel completely at ease interpreting Figure 4. For one thing, what do the negative scores in the upper left subfigure mean?

An interesting but controversial property of this approach is that it tries to map the linguistic embeddings down to a fixed dimension using a linear mapping. This is interesting, but surely the only risk is not *underestimating* the relatedness between tasks. Imagine in the worst case that all information is lost in the mapping, and we always get the zero vector, for two different representations. Surely in that case we should have great performance decoding and thereby *overestimate* the relatedness.

I'll return to my remark that I'm not sure I learn a lot empirically - although, I stress, this paper does have a lot to recommend it, particularly methodologically. I don't really see much in the brain map in Figure 3. The text tells me there's something - not necessarily much - and I guess I trust that, but it does not seem like there's much there. And I have a hard time getting *anything* useful out of Figure 4, aside from the fact that these representations predict voxels. If there is more, and more interpretable, content to be had in these results, I'd encourage the authors to draw it out more. Similarly, the similarity matrix in Figure 2 isn't all that informative, although the MDS is certainly intriguing. However, it leaves a lot of open questions. Is this first component really "semantic depth"? What further tests could we do to probe whether that's the case? How could we operationalize that notion? Second, what is this second dimension doing? And, while the paper suggests that the later layers of some models come close to the next word embedding, I don't see them (visually) coming very close, and a glimpse at the similarity matrix in Figure 2 confirms that for me. The next word embedding is really far from everything. So... doesn't that mean these language models are bad?

In short, I think there is a lot of promising exploration made possible by this paper. I'm not sure there is enough exploration and discussion, as per my comments. And I find the methodology either too complicated, too sparsely motivated, or just hard to follow because it's too dense.

Comments in author review period
===========================

My overall assessment of this paper is exactly the same, except that I realize that the methods (based on Zamir) are slightly less original than I had thought. I am lowering my score to a 7 to reflect this. It is still a very interesting if preliminary exploration.

**Time Spent Reviewing:**

6

---

> ### Author Response · Authors · 2021-08-10
> **Author Response to R1**
>
> Thank you for your thoughtful review! We have responded to your questions in order:
>
> > “Why go to the trouble of the ‘tournament’? Why not just, for each element i, directly use the MSE for decoding the target representation using the representation i? And, for that matter, even in the tournament version, why not take into account the actual values of the MSEs rather than just the percentage for which the one was higher than the other?”
>
> The tournament procedure (which was proposed in the Zamir et al., 2018 taskonomy paper) guarantees that the affinity values are scaled similarly across feature spaces. Some representations are simply harder to predict than others, resulting in different average MSEs as well as different ranges of potential MSEs between representations. We don’t want our representation embedding space to be obfuscated by these difficulty differences, so we used the tournament procedure to correct it. We will clarify this point in a camera-ready version.
>
> > More information would be welcome about what the "temporal transformation" is in 4.2.2.
>
> The “temporal transformation” is simply a time delay commonly used to model the hemodynamic effects present in the BOLD response. We use time delays of 2,4,6, and 8 seconds of the representation to predict the BOLD response. We will add this detail to a camera-ready version.
>
> > In 4.2.3, the methodology is sufficiently complicated that I have a hard time following what exactly \rho is. More importantly, I have a hard time understanding why the \rho values should be sufficiently comparable to the representation embeddings that it would make any sense to project them onto the first basis vector of the MDS.
>
> The \rho values describe how well the responses at each location (voxel) in the brain can be predicted using each language representation. If a certain brain area specialized in processing a specific type of information, such as part of speech, then we would expect representations that capture that information to be good predictors of that brain area, while representations that do not include that information should do poorly. Each voxel in the brain can thus be thought of as another language representation, and we can infer where that representation would lie in the 2-D MDS space by projecting it onto those dimensions. The discriminability analysis (Figure 4) goes further to show that the space of language representations in the brain is—at least partially—isomorphic to the space of artificial language representations. We will clarify these issues in the camera-ready version of this paper.
>
>
> > Similarly, I have a hard time getting the intuitions behind the method in 4.2.4, and, as such, I don't feel completely at ease interpreting Figure 4. For one thing, what do the negative scores in the upper left subfigure mean?
>
> We have added our relevant discussion with R1 about this subject here.
>
> The discriminability score has a somewhat verbose mathematical definition but this is simply to ensure fairness in how we test the representation embeddings. The high-level idea here is that if the representation embeddings actually reflect the structure of how the brain represents linguistic information, then we should be able to “match” each representation embedding to its corresponding performance vector, which describes which regions contain the information in a given representation. We structure this as a leave-two-out experiment, where we train a learner using 98 (representation embedding, performance vector) pairs and then use this learner to try and predict which of the remaining two representation embeddings match with the remaining two performance vectors. If the correlation between the predicted performance vector for representation A matches the ground truth performance vector for representation A better than the ground truth performance vector for representation B (and similarly B matches B better than it matches A), then we have succeeded in correctly discriminating a performance vector from its corresponding representation embedding.
>
> The negative scores represent below-chance performance at discriminating between that pair of representations. This could happen by chance: if two representations are truly not discriminable in the brain, then the discrimination scores should be randomly distributed around zero. But it could also happen if the task affinities are very different from the brain representations. We believe the negative scores for GloVe NWE are caused by the latter. This representation is highly dissimilar from the others in the type of information that it contains, hence its far-removed position in the MDS space. However, the temporal imprecision of the BOLD responses recorded by fMRI (which is on the order of 1 second, or 2-5 words) makes it so that the GloVe NWE and normal GloVe embeddings are functionally very similar for predicting brain data (note that the average prediction performance for NWE and normal GloVe is almost identical in the lower right panel of Figure 4). Because our discrimination test uses the affinities to predict the pattern of fMRI encoding performance across the brain, this disconnect—NWE is different from everything in the affinity analysis, but very similar in the fMRI analysis—is likely causing the negative scores. We will clarify this in the final version of the paper.
>
> > An interesting but controversial property of this approach is that it tries to map the linguistic embeddings down to a fixed dimension using a linear mapping. This is interesting, but surely the only risk is not underestimating the relatedness between tasks. Imagine in the worst case that all information is lost in the mapping, and we always get the zero vector, for two different representations. Surely in that case we should have great performance decoding and thereby overestimate the relatedness.
>
> First, note that the tournament procedure helps ensure that this worst-case scenario cannot happen: if a representation was mapped to all zeros, then the transfer models from every other representation would also easily learn to produce all zeros, and the tournament would assign them all the same (poor) performance (assuming small random fluctuations to break ties) despite low MSE. Second, in practice none of the representations gets mapped to a null space since the autoencoding performance of all the representations to themselves through the dimensionality bottleneck is very good. Very little information is lost in this step.
>
> > Is this first component really "semantic depth"? What further tests could we do to probe whether that's the case? How could we operationalize that notion? Second, what is this second dimension doing?
> These are excellent and interesting questions. Building understanding of this space-of-spaces both qualitatively and quantitatively is the most important next step for this work. However, we hesitate to precisely define how each dimension should be interpreted in isolation. (We do mention that together with the tasks that it tends to separate, the first dimension is evocative of a language hierarchy, but this is simply an observation and not an assertion.)
>
> Instead, it may be more important to consider the entire space, which mostly resides within a triangle that has vertices at the word embeddings, the interpretable spaces (POS, Chunking, Framing, NER), and the next-word embedding. The language models span this triangular space, starting at the word embeddings, arcing toward the interpretable spaces, then arcing toward the next-word embedding. These arcs proceed almost uniformly along the first dimension, from negative (lower LM layers) to positive (higher LM layers). However, on the second dimension the values actually rise and then fall, with the highest values being assigned to the middle layers (e.g. layer 5 of 12 in GPT-2 small). The MDS space suggests that those middle layers are the most similar to the interpretable, human-defined representations (POS, NER, etc.), while the earlier and later layers are less similar. If the human-defined representations are the most abstracted from the input word sequence (i.e. because they are the lowest-dimensional), then this might suggest that the second dimension is capturing something akin to level of abstraction. We will include discussion of this issue in the final version of this paper. We also welcome any suggestions for probes or metrics that can be applied to these representations!
>
>
> > The next word embedding is really far from everything. So... doesn't that mean these language models are bad?
>
> It is very reasonable to believe the next word embedding (NWE) is and should be far from the other representations. Relative to the non-language-model representations, NWE contains information that is fundamentally different, since it is computed from different words. But it also makes sense that NWE is rather far from the LM output representations. This is because the language modelling task is fundamentally uncomputable. If a language model is asked to finish the sentence “The car is ____.”, its hidden state representation must represent the distribution over all possible completions. It can be black/white/big/small, and none of these possibilities are more wrong than the others based on the information available to the language model. So the LM representations are going to include information about that uncertainty, whereas the NWE representation will not, as it has special oracle access to this uncomputable ground truth.

---

### Decision · Program_Chairs · 2021-09-27

**Decision:**

Accept (Poster)

**Comment:**

Reviewers found that the method presented in this submission is useful and casts our understanding of language processing in the brain in a new light.

I encourage the authors to update their manuscript to include the results from the discussion with the reviewers, as these were decisive.